# Transcriptional programs regulating neuronal differentiation are disrupted in DLG2 knockout human embryonic stem cells and enriched for schizophrenia and related disorders risk variants

Bret Sanders[1], Daniel D'Andrea [2], Mark O. Collins [3], Elliott Rees[2], Tom G. J. Steward [4], Ying Zhu[1], Gareth Chapman[1], Sophie E. Legge [2], Antonio F. Pardiñas [2], Adrian J. Harwood[1], William P. Gray [1], Michael C. O'Donovan [2], Michael J. Owen [1,2], Adam C. Errington[1], Derek J. Blake [2], Daniel J. Whitcomb [4], Andrew J. Pocklington [2✉] & Eunju Shin [1,5✉]

Coordinated programs of gene expression drive brain development. It is unclear which transcriptional programs, in which cell-types, are affected in neuropsychiatric disorders such as schizophrenia. Here we integrate human genetics with transcriptomic data from differentiation of human embryonic stem cells into cortical excitatory neurons. We identify transcriptional programs expressed during early neurogenesis in vitro and in human foetal cortex that are down-regulated in $DLG2^{-/-}$ lines. Down-regulation impacted neuronal differentiation and maturation, impairing migration, morphology and action potential generation. Genetic variation in these programs is associated with neuropsychiatric disorders and cognitive function, with associated variants predominantly concentrated in loss-of-function intolerant genes. Neurogenic programs also overlap schizophrenia GWAS enrichment previously identified in mature excitatory neurons, suggesting that pathways active during prenatal cortical development may also be associated with mature neuronal dysfunction. Our data from human embryonic stem cells, when combined with analysis of available foetal cortical gene expression data, de novo rare variants and GWAS statistics for neuropsychiatric disorders and cognition, reveal a convergence on transcriptional programs regulating excitatory cortical neurogenesis.

[1] Neuroscience and Mental Health Research Institute, Cardiff University, Cardiff CF24 4HQ, UK. [2] MRC Centre for Neuropsychiatric Genetics and Genomics, Cardiff University, Cardiff CF24 4HQ, UK. [3] Department of Biomedical Science, University of Sheffield, Sheffield S10 2TN, UK. [4] Bristol Medical School, University of Bristol, Bristol BS1 3NY, UK. [5] School of Life Sciences, Keele University, Keele ST5 5BG, UK. ✉email: pocklingtonaj@cardiff.ac.uk; e.shin@keele.ac.uk

Schizophrenia (SZ) is a highly heritable[1,2] psychiatric disorder, with genetic variation ranging from common polymorphisms (SNPs) to rare mutations contributing to disease risk[3–7]. Rare variant studies consistently implicate disruption of postsynaptic signaling complexes in SZ etiology[4,5,8–11], however the cellular pathways mediating common variant risk (an estimated 30–50% of the total genetic contribution to liability[5]) remain unclear. Genome-wide association studies (GWAS) have shown SZ common variant enrichment in broad, synapse-related gene sets[12,13], but these sets only capture a modest proportion of the overall common variant association signal[12]. In contrast, nearly 50% of genic SNP-based heritability is captured by loss-of-function intolerant (LoFi) genes[12]. Being under extreme selective constraint, LoFi genes are likely to play important developmental roles. Indeed, LoFi genes are enriched for rare variants contributing to autism spectrum disorders (ASD) and intellectual disability/severe neurodevelopmental delay (ID/NDD)[14], conditions that manifest early in life. Rare variation in LoFi genes, including many of those implicated in ASD and ID/NDD, also contributes to SZ[11,15,16]. We hypothesized that a significant proportion of SZ common variants may contribute to disease via the disruption of neurodevelopmental pathways harboring a concentration of LoFi genes.

Supporting a neurodevelopmental role for SZ common variants, there is growing evidence that many such risk factors impact gene expression in the fetal brain[17–20] and are enriched in cell-types at multiple stages of cortical excitatory neuron development[21]. This raises the question: do SZ common variants converge on specific gene expression programs that are normally activated or repressed during fetal cortical excitatory neuron development? Mutations disrupting regulators of such programs would be expected to have a larger effect size and lower allele frequency than risk variants impacting individual genes within the program. We therefore sought rare, single-gene mutations linked to SZ where the affected gene is expressed in human fetal brain and has the potential to regulate developmental processes. This led us to DLG2. Firstly, multiple independent deletions have been identified at the DLG2 locus in SZ and ASD patients[8,22]. Secondly, DLG2 mRNA is present from 8 weeks post-conception in humans[23] and throughout all stages of in vitro differentiation from human embryonic stem cells (hESCs) to cortical projection neurons[24]. Thirdly, suggesting a potential regulatory role for DLG2 during early development, the single invertebrate orthologue of DLG1-4 (Dlg) is a core component of the Scrib signaling module, which regulates cell polarity, differentiation and migration during development[25]. Primarily studied as a post-synaptic scaffold protein, DLG2 is required for the formation of NMDA receptor complexes[26]. These complexes regulate the induction of several forms of synaptic plasticity[27] and are enriched for rare mutations in SZ cases[4,5,8–11]. This raises the possibility that DLG2 may be required for the normal operation of both adult and developmental signaling pathways relevant to SZ pathophysiology.

To explore the role of DLG2 in neurodevelopment we engineered hESCs with homozygous loss-of-function DLG2 mutations ($DLG2^{-/-}$) using the CRISPR-CAS9 system. $DLG2^{-/-}$ knockout (KO) and isogenic sister wild-type (WT) hESC lines were differentiated into cortical excitatory neurons and cells were characterized at multiple developmental timepoints to identify phenotypes and gene expression changes in KO lines (Fig. 1a). Neurodevelopmental gene expression programs dysregulated in $DLG2^{-/-}$ lines were identified and analyzed for risk variant enrichment, first for SZ and then for related disorders. We explored the biological function of disease-associated programs, both computationally and experimentally, and evaluated the contribution of LoFi genes to common and rare variant associations. Returning to SZ, we investigated the relationship between developmental and mature neuronal pathways enriched for common variant association. Finally, we explored whether disease-associated neurogenic programs identified in vitro possessed a similar profile of expression across neurodevelopmental cell-types in human fetal cortex.

## Results

**Knockout generation and validation.** DLG2 contains three PDZ domains, an SH3 and a GK domain, all involved in protein binding. Two $DLG2^{-/-}$ lines were created from H7 hESCs using the CRISPR/Cas9-D10A nickase system targeting the first PDZ domain, generating a frameshift and premature stop codon in both alleles (Supplementary Fig. 1). Sequencing of predicted off-target sites revealed no mutations (Methods, Supplementary Fig. 2, Supplementary Data 1). All subsequent analyses compared these lines to an isogenic WT sister line that underwent the same procedure but remained genetically unaltered.

$DLG2^{-/-}$ and WT lines were differentiated into cortical excitatory neurons using a modified dual SMAD inhibition protocol[28,29]; RNA was extracted in triplicate from each line at 4 timepoints spanning cortical excitatory neuron development and gene expression quantified (Fig. 1b, Supplementary Fig. 3). A significant decrease in DLG2 mRNA was observed for exons and transcripts spanning the first PDZ domain, indicating degradation via nonsense-mediated decay in $DLG2^{-/-}$ lines (Supplementary Fig. 4). Quantitative mass spectrometry of peptide-affinity pulldowns using the NMDA receptor NR2 subunit PDZ peptide ligand[30] identified DLG2 in WT only, confirming the absence of DLG2 in KO lines (Fig. 1c–f, Supplementary Data 2). Genotyping revealed no CNVs in either $DLG2^{-/-}$ line relative to WT (Supplementary Fig. 5a). Both $DLG2^{-/-}$ lines expressed pluripotency markers OCT4, SOX2 and NANOG at 100% of WT levels (Supplementary Fig. 5b). Cells were extensively characterized for their cortical identity using western blotting and immunocytochemistry from days 20–60. Over 90% of day 20 cells were positive for FOXG1, PAX6 and SOX2 and <1% cells expressed ventral genes such as DLX1, GBX2, NKX2.1 and OLIG3 (Supplementary Fig. 6), confirming dorsal forebrain fate. In addition, staining of markers expressed in ventral forebrain-derived neurons from striatal, thalamic and hypothalamic nuclei confirmed no or trace expression (Supplementary Fig. 6).

**$DLG2^{-/-}$ alters gene expression during cortical differentiation.** To robustly identify genes dysregulated by DLG2 knockout, expression data from the two $DLG2^{-/-}$ lines was pooled and compared to WT at each timepoint (Methods). Disruption of DLG2 had a profound effect: of the >13,000 protein-coding genes expressed at each timepoint, ~7% displayed altered expression at day 15, rising to 40–60% between days 20 and 30 then decreasing to ~25% by day 60 (Fig. 1g, Supplementary Data 3).

**Common risk variants implicate disruption of neurogenesis in SZ.** We next tested whether genes differentially expressed in $DLG2^{-/-}$ lines at each timepoint were enriched for SZ common risk variants. Taking summary statistics from the most recent SZ GWAS available[12], we utilized the competitive gene-set enrichment test implemented in MAGMA[31]. As expected for cells of neural lineage, the set of all genes expressed at one or more timepoint in $DLG2^{-/-}$ or WT lines ($all^{WT+KO}$) was enriched for common variant association ($P = 8.03 \times 10^{-21}$, $N_{gene} = 14,274$). To investigate whether genes up-/down-regulated at each timepoint displayed SZ association above that seen for neurodevelopmentally expressed genes in general, we tested them for association conditioning on $all^{WT+KO}$. This revealed enrichment solely for genes down-regulated at day 30

$(30_\text{down}^{-/-}: P_\text{corrected} = 9.5 \times 10^{-8}$ Fig. 2a), coinciding with active neurogenesis (Fig. 1b). Conditioning on timepoint-specific expressed genes gave the same result (Supplementary Data 4).

Compared to $all^{WT+KO}$, $30_\text{down}^{-/-}$ genes were over-represented in Gene Ontology (GO) terms related to neuronal development, function and migration (Methods, Supplementary Data 5). Iterative refinement via conditional analyses identified 23 terms with independent evidence for over-representation (Fig. 2b, Methods). This suggests that loss of *DLG2* dysregulates transcriptional programs underlying neurogenesis (neuronal differentiation, migration and maturation) and implicates these processes in SZ etiology.

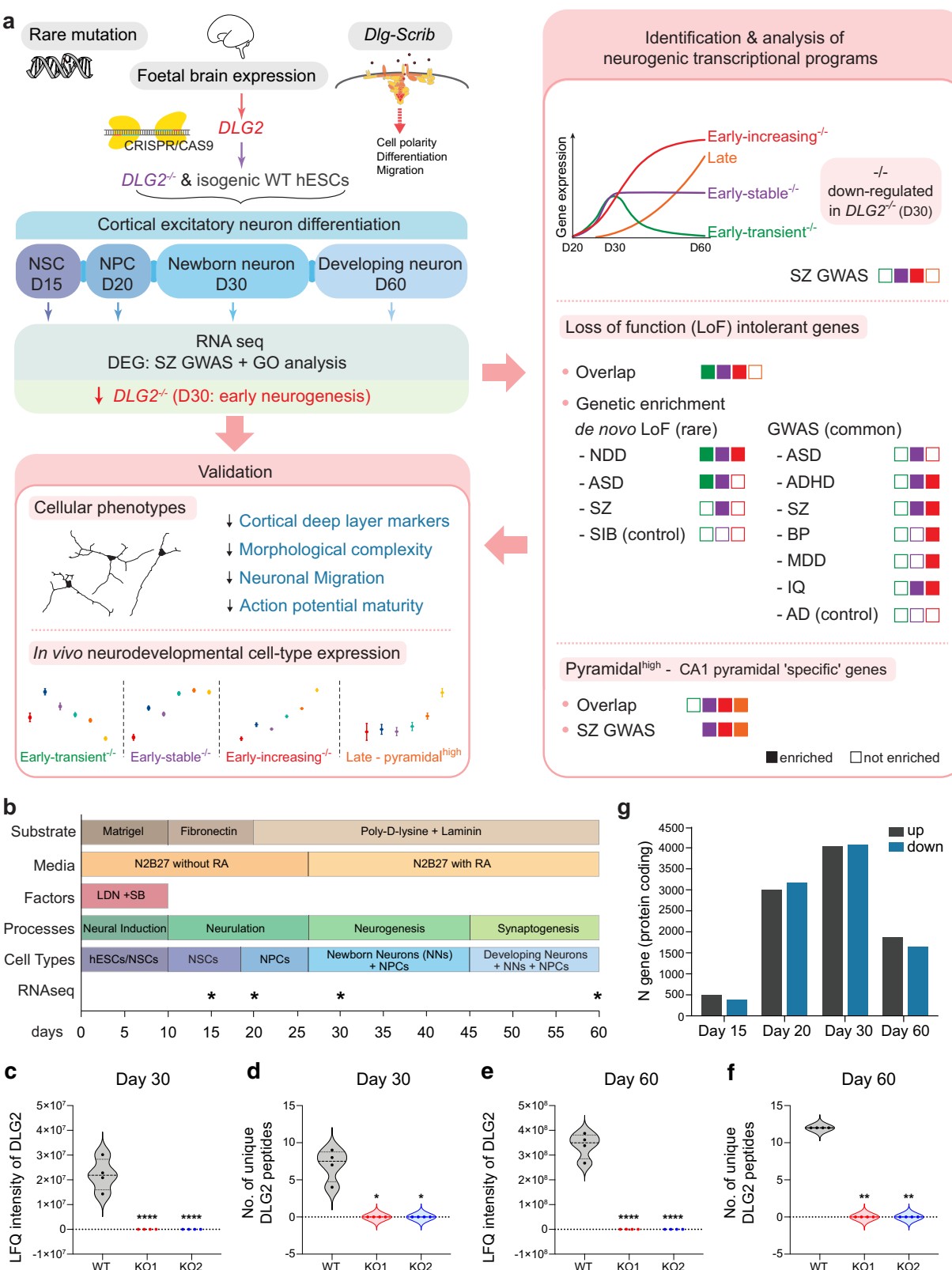

**Fig. 1 Study design and number of differentially expressed genes. a** Study summary. $DLG2^{-/-}$ and wild-type (WT) hESCs were differentiated into cortical excitatory neurons and RNA collected at multiple timepoints: predominant cell types shown for each timepoint (NSCs, neural stem cells; NPCs, neural precursor cells). Genetic analysis of differentially expressed genes (DEG) revealed SZ GWAS enrichment in genes down-regulated at day 30 in $DLG2^{-/-}$ lines, coinciding with early neurogenesis: corresponding phenotypes predicted via GO term analysis were validated experimentally. Transcriptional programs active during neurogenesis were identified based on differential gene expression between successive developmental timepoints. Schizophrenia (SZ) common variant risk was concentrated in two early neurogenic programs down-regulated in $DLG2^{-/-}$ cells. Loss of function intolerant (LoFi) genes were over-represented in early neurogenic (but not late) programs. LoFi genes in early neurogenic programs were enriched for common/rare variants contributing to mental disorders (ASD Autism spectrum disorder; ADHD attention-deficit hyperactivity disorder; BP Bipolar disorder; MDD major depression disorder) and cognition (IQ), but not unaffected siblings of ASD cases (SIB) or Alzheimer's disease (AD). Overlap with early and late neurogenic programs captures SZ GWAS association previously reported for genes with high expression in CA1 pyramidal neurons relative to other brain cell-types (pyramidal[high]). The expression profile seen for each disease-associated neurogenic program in vitro was recapitulated across neurodevelopmental cell-types from human fetal cortex. **b** Overview of cortical differentiation protocol with approximate timings of key developmental processes and predominant cell types present in culture. Asterisks indicate timepoints selected for RNA sequencing. **c–f** Label free quantification (LFQ) of DLG2 protein levels in PDZ-ligand (NR2 C-terminus) affinity pulldowns ($n = 4$ biological repeats each) in day 30 and 60 WT and $DLG2^{-/-}$ cells using LC-MS/MS analysis. One-way ANOVA with Bonferroni multiple comparison correction applied to **c** ($F_{2,9} = 45.54$, $P = 1.96 \times 10^{-5}$) and **e** ($F_{2,9} = 172.9$, $P = 6.59 \times 10^{-8}$). Kruskal–Wallis test with Dunn's multiple comparison correction applied to **d** (H(2) = 10.46, $p = 0.0061$) and **f** (H(2) = 11.00, $p = 0.0061$). *$p < 0.05$; **$p < 0.01$; ****$p < 0.0001$ vs. WT. **g** Number of protein coding genes differentially expressed in $DLG2^{-/-}$ cells relative to WT at each timepoint. Source data are provided as a Source Data file.

**$DLG2^{-/-}$ delays cortical cell-fate expression in newborn neurons.** To validate disruption of neurogenesis in $DLG2^{-/-}$ lines and investigate whether this leads to differences in the number or type of neurons produced, we compared the expression of cell-type specific markers in $DLG2^{-/-}$ and WT lines from days 30–60 via immunocytochemistry (ICC) and Western blotting (Fig. 2c–i). From ICC it was clear that $DLG2^{-/-}$ cells are able to differentiate and produce postmitotic neurons expressing characteristic neuronal markers such as NEUN and TUJ1 plus cortical deep layer markers TBR1 and BCL11B (CTIP2) (Fig. 2c–i, Supplementary Fig. 7). Western blot of NEUN (Fig. 2c) and MAP2 (Supplementary Fig. 7) and quantification of NEUN+ cells following ICC (Fig. 2f) revealed no difference in the percentage of neurons produced by $DLG2^{-/-}$ cultures. This is in line with the comparable percentage of cells in the cell cycle/neural progenitors at days 30–60 in $DLG2^{-/-}$ and WT cultures indicated by a similar proportion of KI67+ and SOX2+ cells (Supplementary Fig. 7). At these early timepoints we would not expect to see the generation of upper layer neurons, which express markers such as SATB2. Although we could identify a small percentage of SATB2+ cells in both WT and KO lines, all co-expressed CTIP2 (Supplementary Fig. 7) indicating their deep layer identity[32]. An analysis of deep layer markers TBR1 and CTIP2 revealed a significant decrease in CTIP2+ cells but a comparable proportion of TBR1+ neurons for all timepoints investigated (Fig. 2d, e, g–i). On average the proportion of CTIP2+ cells recovered from 15% of the WT level on day 30 to 50% by day 60, although there was notable variation between $DLG2^{-/-}$ lines (Supplementary Fig. 8); total CTIP2 protein levels also recovered to some extent, but at a slower rate (Supplementary Fig. 8). Thus, $DLG2^{-/-}$ does not affect the rate at which neurons are produced but delays the expression of subtype identity in newborn deep layer neurons.

**$DLG2^{-/-}$ lines display deficits in neuron morphology & migration.** Given the over-representation of $30_{down}^{-/-}$ genes in terms related to neuron morphogenesis and migration (Fig. 2b), we sought to experimentally validate these phenotypes. Immature (day 30) and mature (day 70) neurons were traced and their morphology quantified (Fig. 3). At both timepoints $DLG2^{-/-}$ neurons displayed a simpler structure than WT, characterized by a similar number of primary neurites projecting from the soma (Fig. 3a) but with greatly reduced secondary neurite branching

(Fig. 3b). Total neurite length did not differ (Fig. 3c), leading to a clear $DLG2^{-/-}$ phenotype of longer, relatively unbranched primary neurites (Fig. 3e). There was no significant difference in soma area (Fig. 3d). Day 40 $DLG2^{-/-}$ neurons had a slower speed of migration (Fig. 3f) and reduced displacement from their origin after 70 h (Fig. 3g, h). In summary, $DLG2^{-/-}$ neurons show clear abnormalities in both morphology and migration, validating the GO term analysis.

**DLG2-regulated transcriptional programs enriched for SZ genetic risk.** We postulated that loss of $DLG2$ inhibits the activation of transcriptional programs driving neurogenesis, which starts between days 20 and 30 and steadily increases thereafter. If this is the case, SZ association in $30_{down}^{-/-}$ should be captured by genes normally upregulated between days 20 and 30 in WT cultures ($20$–$30_{up}^{WT}$). Analyzing differential expression between WT samples at successive timepoints, we found risk variant enrichment only in $20$–$30_{up}^{WT}$ (conditioning on all WT-expressed genes, Fig. 4a). Most $20$–$30_{up}^{WT}$ genes overlapped $30_{down}^{-/-}$ (3075 genes, 85%) and this overlap captured the signal in both sets ($P_{overlap} = 3.23 \times 10^{-10}$; $30_{down}^{-/-}$ only $P = 0.44$; $20$–$30_{up}^{WT}$ only $P = 0.62$). This was not simply due to the size of the overlap as the regression coefficient for this set ($\beta = 0.14$), which reflects magnitude of enrichment, was significantly greater than for genes unique to $30_{down}^{-/-}$ ($\beta = 0.006$, $P_{greater} = 0.00077$) or $20$–$30_{up}^{WT}$ ($\beta = -0.015$, $P_{greater} = 0.0023$). Thus, it is neurogenic transcriptional programs that are typically upregulated in WT but down-regulated in $DLG2^{-/-}$ lines that are enriched for SZ common variants.

To more precisely identify SZ-associated transcriptional programs active during neurogenesis, we classified $20$–$30_{up}^{WT}$ genes based on their subsequent WT expression profiles (Fig. 4b, Methods). These included early-increasing genes, whose expression continues to rise between days 30 and 60; early-stable genes, whose expression stays at a relatively constant level; and early-transient genes, whose expression is later down-regulated. We also defined a set of late genes, whose expression only increases significantly after day 30. These were further partitioned into genes that were down-regulated at day 30 in $DLG2^{-/-}$ lines (e.g., early-stable$^{-/-}$) and those that were not (e.g., early-stable$^{WT\ only}$). The sole exception to this was the late set, which had minimal overlap with $30_{down}^{-/-}$ (62 out of 1399 genes) and was therefore left intact.

Early-stable$^{-/-}$ and early-increasing$^{-/-}$ sets were robustly enriched for SZ association (Fig. 4c). To more precisely control for association specifically in neuron-expressed genes, we identified genes expressed in newborn and developing cortical excitatory neurons from a recent single-cell RNAseq study of human fetal brain tissue[32] (Methods). Early-stable$^{-/-}$ and early-increasing$^{-/-}$

remained highly associated when conditioning on fetal neuron-expressed genes (Supplementary Data 6). Furthermore, $all^{WT+KO}$ displayed association that was not captured by fetal neuron-expressed genes. We therefore continued to condition genetic analyses on $all^{WT+KO}$, as this best captures the broad SZ signal from neuronal-lineage genes present in our dataset.

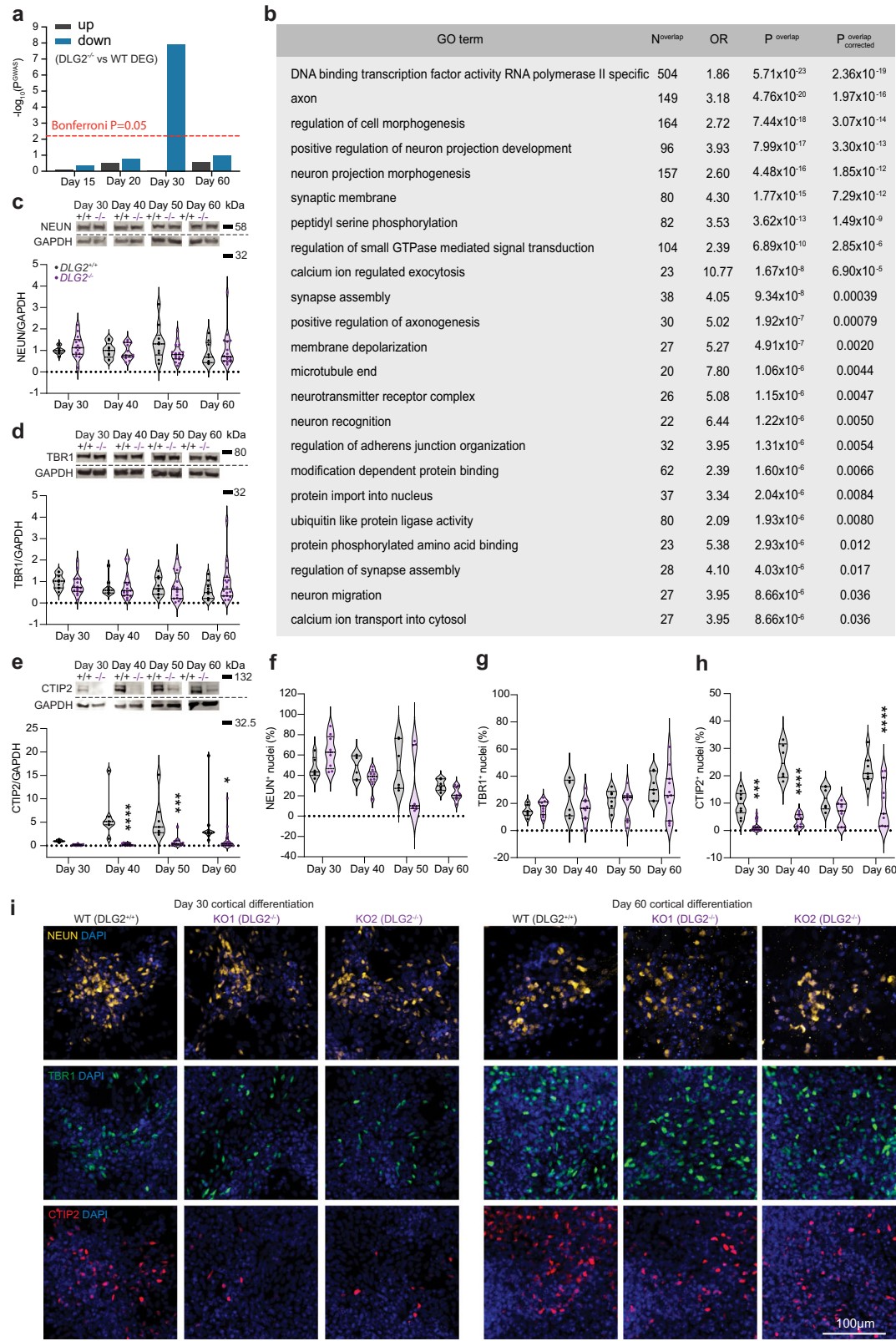

**Fig. 2 Common risk variants implicate disruption of neurogenesis in schizophrenia. a** Schizophrenia GWAS association in genes up-/down-regulated (DEG) at each timepoint ($DLG2^{-/-}$ relative to WT), conditioning on all expressed genes. **b** Gene ontology (GO) terms over-represented amongst genes down-regulated at day 30 in $DLG2^{-/-}$ lines relative to all expressed genes (one-sided Fisher's Exact Test). Number of overlapping genes ($N^{overlap}$), enrichment odds ratio (OR), plus raw and Bonferroni-corrected $p$ values are given ($p^{overlap}$, $p^{overlap}_{corrected}$), where correction is for the total number of GO terms tested. **c** NEUN western blot protein quantification. Neither genotype ($F_{1,90} = 0.1852$; $P = 0.6680$) nor time ($F_{3,90} = 0.5382$; $P = 0.6573$) had significant effects on NEUN expression (WT $n = 10, 9, 10, 9$; KO $n = 14, 14, 16, 16$ biological repeats from 2 independent experiments for each genotype, days 30, 40, 50 & 60 respectively). **d** TBR1 western blot protein quantification. Neither genotype ($F_{1,95} = 0.3899$; $P = 0.5338$) nor time ($F_{3,35} = 0.5052$; $P = 0.6793$) had significant effects on TBR1 expression (WT $n = 10, 10, 9, 10$ biological repeats, days 30, 40, 50 & 60 respectively; KO $n = 16$ biological repeats at each timepoint; from 3 independent experiments for each genotype). **e** CTIP2 western blot protein quantification. Genotype ($F_{1,86} = 39.89$; $P = 1.14 \times 10^{-8}$) and time ($F_{3,86} = 5.262$; $P = 0.0022$) had significant effects on CTIP2 expression (WT $n = 10, 7, 7, 10$; KO $n = 17, 12, 13, 18$ biological repeats; from 2 independent experiments for each genotype, days 30, 40, 50 & 60 respectively). **f** ICC quantification of NEUN$^+$ cells. Time ($F_{3,52} = 7.018$, $P = 0.0005$) had a significant effect on NEUN expression; genotype ($F_{1,52} = 1.687$; $P = 0.1998$) did not (WT $n = 6$ biological repeats; KO $n = 9$ biological repeats from 2 independent experiments at each timepoint for each genotype). **g** ICC quantification of TBR1$^+$ cells. Time ($F_{3,58} = 4.738$, $P = 0.0050$) had a significant effect on TBR1 expression; genotype ($F_{1,58} = 1.664$; $P = 0.2022$) did not (WT $n = 7, 6, 6, 7$; KO $n = 11, 9, 9, 11$ biological repeats from 3, 2, 2, 3 independent experiments for each genotype, days 30, 40, 50 & 60 respectively). **h** ICC quantification of CTIP2$^+$ cells. Genotype ($F_{1,67} = 101.8$; $P = 4.46 \times 10^{-15}$) and time ($F_{3,67} = 18.93$; $P = 5.33 \times 10^{-9}$) had significant effects on CTIP2 expression (WT $n = 10, 6, 6, 7$; KO $n = 17, 9, 9, 11$ biological repeats from 3, 2, 2, 3 independent experiments for each genotype, days 30, 40, 50 & 60 respectively). **i** Representative ICC images of NEUN, TBR1 and CTIP2 with DAPI nuclear counterstain. Western blotting (**c–e**) and ICC data (**f–h**) were analyzed by two-way ANOVA with post hoc comparisons using Bonferroni correction, comparing to WT controls. Stars indicate Bonferroni-corrected $p$ values, *$P < 0.05$; **$P < 0.01$; ***$P < 0.001$; ****$P < 0.0001$ vs. WT control. Source data are provided as a Source Data file.

In summary, SZ GWAS association during early neurogenesis is restricted to 2 transcriptional programs down-regulated in $DLG2^{-/-}$ lines.

**Transcriptional cascade predicted to drive early neurogenesis**. We next investigated the biological function of early neurogenic programs dysregulated in $DLG2^{-/-}$ lines. Each was over-represented for a coherent set of GO terms indicating a distinct biological role (Supplementary Data 7): early-transient$^{-/-}$ for histone/chromatin binding and transcriptional regulation; early-stable$^{-/-}$ for signal transduction, transcriptional regulation, neurogenesis, cell projection development, migration and differentiation; and early-increasing$^{-/-}$ for axon guidance, dendrite morphology, components of pre- and post-synaptic compartments and electrophysiological properties. These functions suggest a linked, time-ordered cascade of transcriptional programs spanning early neurogenesis. This begins with an initial phase of chromatin remodeling (early-transient$^{-/-}$) that establishes neuron sub-type identity and leads to activation of a longer-term program guiding the growth and migration of newborn neurons (early-stable$^{-/-}$). This in turn promotes the development and fine-tuning of sub-type specific neuronal structure, function and connectivity as cells enter the terminal phase of differentiation (early-increasing$^{-/-}$).

To test support for the existence of such a cascade and its disruption in disease, we identified disease-associated regulatory genes from each program whose downstream targets have been experimentally identified or computationally predicted (Methods). Reflecting our hypothesis that dysregulation of these pathways is likely to play a role in multiple neurodevelopmental disorders, we sought regulators linked to SZ, ASD and ID/NDD. This led us to chromatin modifier CHD8[33–37] from early-transient$^{-/-}$; transcription factor TCF4[12,38–40] and translational regulator FMRP[12,41,42] from early-stable$^{-/-}$; and transcription factors (and deep layer markers) TBR1[42,43] and BCL11B (CTIP2)[12,13,42,44] from early-increasing$^{-/-}$.

To link successive phases of the cascade, we predicted that each program would be enriched for direct targets of regulators in the immediately preceding program: early-stable$^{-/-}$ for CHD8 targets, early-increasing$^{-/-}$ for TCF4 and FMRP. TBR1 and BCL11B play important roles in the developmental

expression of sub-type specific properties and would be predicted to regulate early-increasing$^{-/-}$ genes. Since early-transient$^{-/-}$ is hypothesized to initiate the cascade, we predicted that early-increasing$^{-/-}$ would be enriched for indirect targets of CHD8 – genes not directly regulated but whose expression is down-regulated in $CHD8$ knockdown cells[36]. We also predicted that genes in the earliest, most transitory phase of the cascade (i.e., early-transient$^{-/-}$) would not be enriched for targets of terminal phase regulators (BCL11B and TBR1). FMRP represses the translation of its mRNA targets, facilitating their translocation to distal sites of protein synthesis[41,45], and its function is known to be important for axon and dendrite growth[46]. We therefore predicted that early-stable$^{-/-}$ (but not early-transient$^{-/-}$) would also be enriched for FMRP targets. Over-representation tests confirmed these predictions, supporting the existence of a regulatory cascade driving early neurogenic programs disrupted in neuropsychiatric disorders (Fig. 4d). In addition, the targets of TCF4, FMRP, BCL11B and TBR1 were more highly enriched for SZ association than other genes in early-increasing$^{-/-}$ (Fig. 4d), highlighting specific pathways through which these known risk genes may contribute to disease.

**Convergence of genetic risk on perturbed action potential generation**. We next tested whether biological processes over-represented in early-stable$^{-/-}$ or early-increasing$^{-/-}$ (Supplementary Data 7) captured more or less of the SZ association in these programs than expected (Methods). Iterative refinement identified 13 GO terms with independent evidence for over-representation in early-stable$^{-/-}$. Genetic association for these terms did not differ substantially from early-stable$^{-/-}$ as a whole (Supplementary Data 8), indicating that risk factors are distributed relatively evenly between them. None of the 16 independent terms identified for early-increasing$^{-/-}$ showed evidence for depleted association, suggesting that diverse biological processes regulating neuronal differentiation, morphology and function are perturbed in SZ. However, *somatodendritic compartment* and *membrane depolarization during action potential* were more highly associated than early-increasing$^{-/-}$ as a whole (Fig. 4e). Enhanced enrichment in action potential (AP) related genes is noteworthy: while postsynaptic complexes

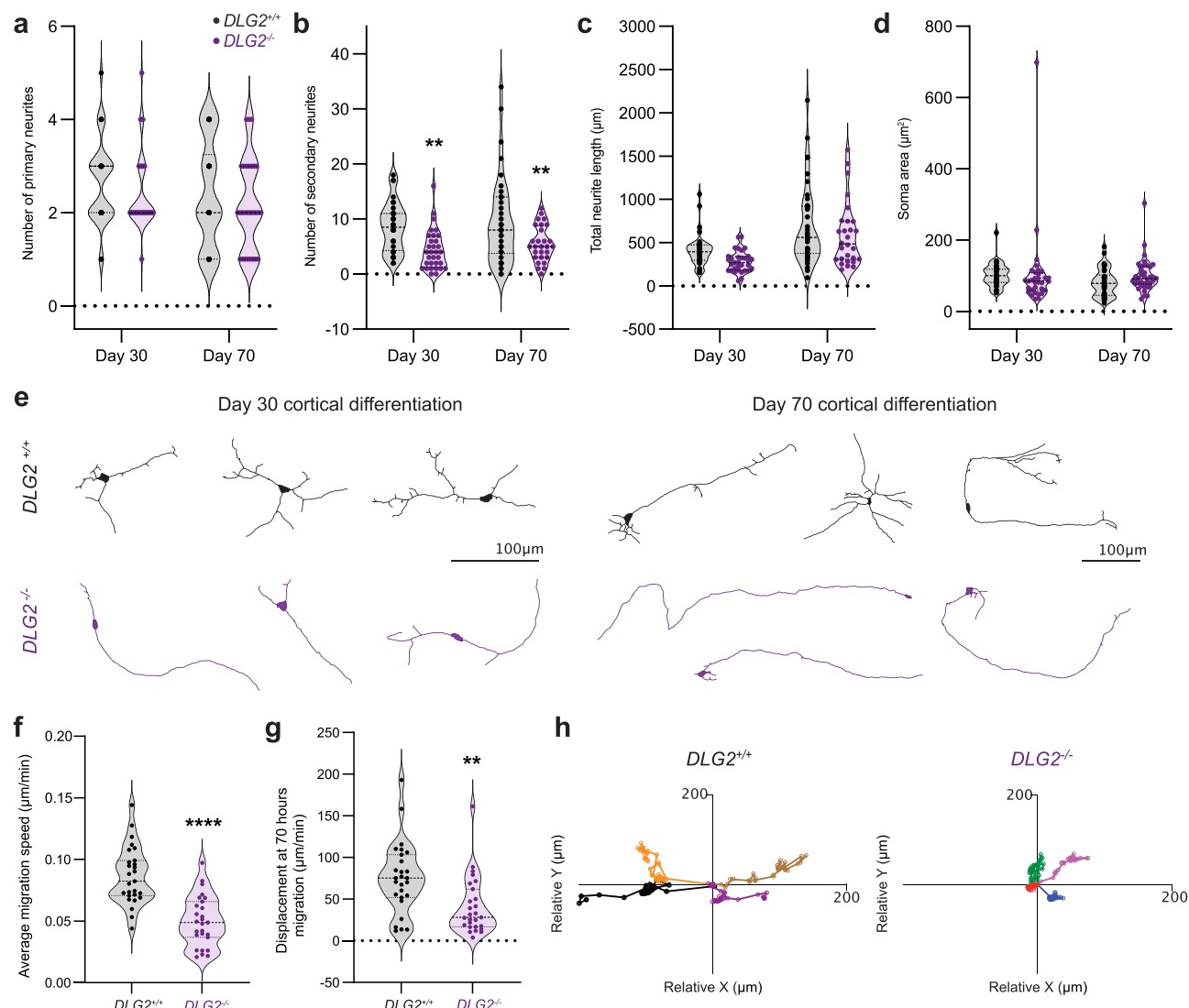

**Fig. 3 $DLG2^{-/-}$ lines display deficits in neuron morphology & migration. a** The number of primary neurites (projecting from the soma). Neither genotype ($F_{1,126} = 1.591$; $P = 0.2095$) nor time ($F_{1,126} = 2.278$; $P = 0.1337$) had significant effects the numbers of primary neurites. **b** The number of secondary neurites (projecting from primary neurites). Genotype ($F_{1,126} = 18.78$, $P = 2.97 \times 10^{-5}$) had a significant effect on number of secondary neurites, while time ($F_{1,126} = 1.082$, $P = 0.3003$) did not. **c** The total neurite length. Both genotype ($F_{1,126} = 4.568$; $P = 0.0345$) and time ($F_{1,126} = 26.33$; $P = 1.06 \times 10^{-6}$) had significant effects on total neurite length. However, post hoc analysis showed no significant differences at individual timepoints. **d** The soma area. Neither genotype ($F_{1,136} = $ ; $P = 0.9170$) nor time ($F_{1,136} = 1.399$; $P = 0.2390$) had a significant effect on soma area. For **a–d** WT $n = 32$, 38 cells from 3 independent experiments, days 30 & 70 respectively; KO $n = 32$, 28 cells from 3 independent experiments, days 30 & 70 respectively. **e** Representative traces showing the neuronal morphology. **f** The average speed of neuronal migration over 70 h, from day 40 of cortical differentiation. $DLG2^{-/-}$ neurons showed significantly decreased average migration speed compared to WT ($t_{52} = 6.1175$; $P = 1.26 \times 10^{-7}$; $n = 27$). For **f–g** both WT & KO $n = 27$ cells from 3 independent experiments. **g** The displacement of neurons at 70 h migration. $DLG2^{-/-}$ neurons showed significantly decreased displacement compared to WT ($t_{52} = 3.244$; $P = 0.0021$; $n = 27$). **h** Representative traces of neuronal migration from a given origin over 70 h. Morphology data sets (**a–d**) were analyzed by two-way ANOVA with post hoc comparisons using Bonferroni correction, comparing to WT controls. Migration data sets (**f**, **g**) were analyzed by unpaired two-tailed Student's $t$ test. Stars above bars represent, $**P < 0.01$; $****P < 0.0001$ vs. WT control (Bonferroni-corrected for morphology analyses). Source data are provided as a Source Data file.

regulating synaptic plasticity are robustly implicated in SZ[4,5,8–11], this suggests that the molecular machinery underlying AP generation is also disrupted. We therefore sought to confirm the disruption of APs in $DLG2^{-/-}$ lines (Fig. 5a–j), also investigating the impact of $DLG2$ loss on synaptic transmission (Fig. 5l–n).

In line with the above, $DLG2^{-/-}$ neurons were found to be less excitable, with immature AP waveforms. Day 50 $DLG2^{-/-}$ neurons displayed a significantly more depolarized resting membrane potential (Fig. 5a). Upon stepped current injection, 80% of WT neurons but only 43% of $DLG2^{-/-}$ neurons showed AP firing (Fig. 5c). APs produced by $DLG2^{-/-}$ neurons were characteristic of less mature neurons (Fig. 5d), having smaller amplitude, longer half-width and a slower maximum rate of depolarization and repolarisation ($\delta V/\delta t$) (Fig. 5e–h). We found no change in AP voltage threshold, rheobase current (Fig. 5i, j) or input resistance (Fig. 5b). In addition, the percentage of neurons displaying spontaneous excitatory postsynaptic currents (EPSCs)

was comparable at days 50 and 60 (Fig. 5n) as was EPSC frequency and amplitude (Fig. 5l, m). Lack of effect on synaptic transmission may reflect compensation by DLG4, whose expression shows a trend towards an increase in synaptosomes from day 65 $DLG2^{-/-}$ neurons (Fig. 5o). In summary, developing $DLG2^{-/-}$ neurons have a reduced ability to fire and produce less mature APs.

**Neurogenic programs capture genetic association in LoFi genes.** Having identified neurodevelopmentally expressed pathways enriched for common SZ risk variants and investigated the phenotypic consequences of their dysregulation in $DLG2^{-/-}$ lines, we sought to test our hypothesis that these pathways capture a significant proportion of the SZ GWAS enrichment seen in

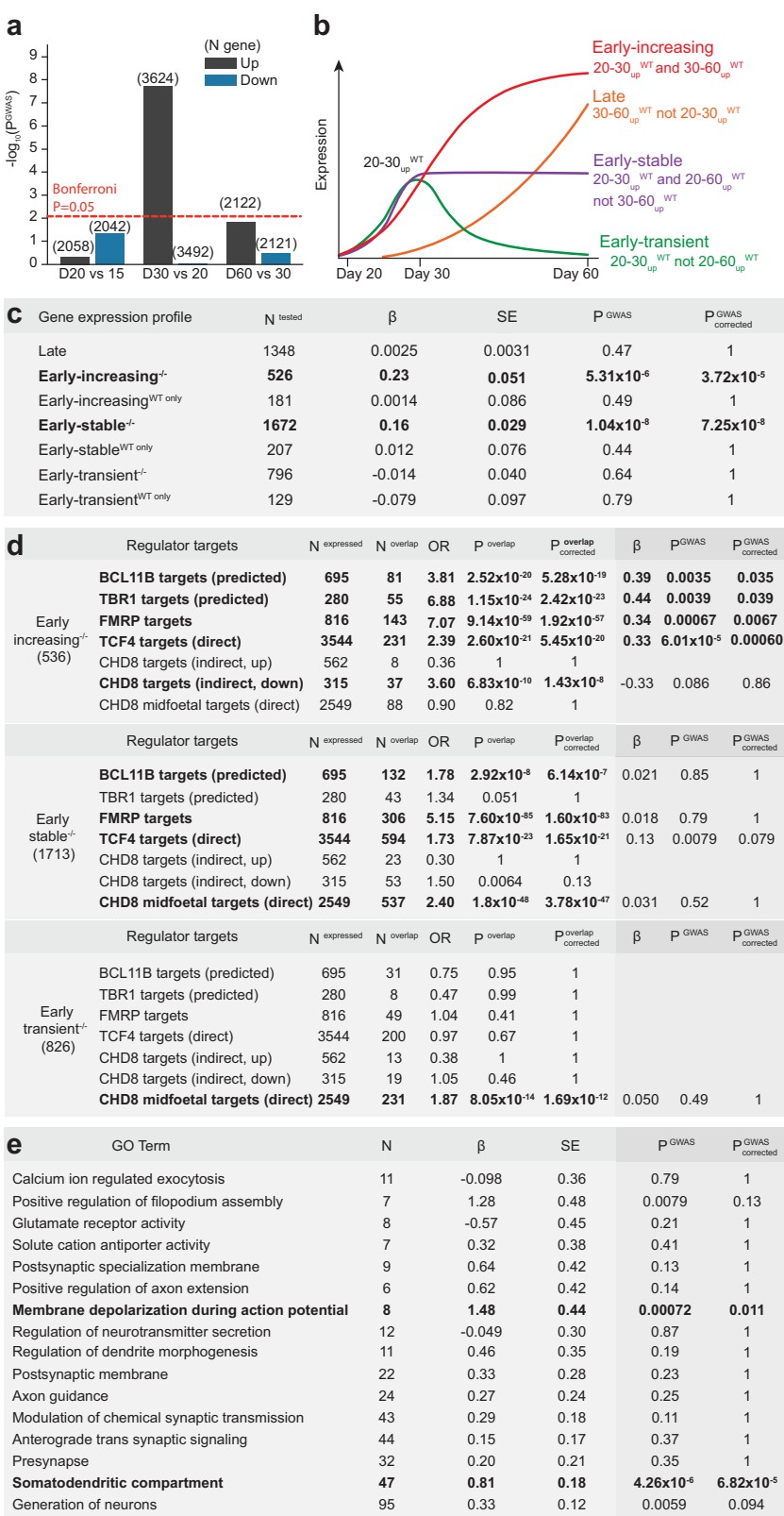

**Fig. 4 DLG2 regulates a cascade of transcriptional programs driving neurogenesis & differentiation. a** Enrichment for common schizophrenia risk variants in genes up- & down-regulated between each successive pair of timepoints in wild-type (WT), conditioning on all WT-expressed genes. Dotted line indicates $P_{corrected} = 0.05$ following Bonferroni correction for 6 tests. **b** Four discrete transcriptional programs initiated following the onset of neurogenesis were identified based upon WT differential expression between timepoints: *early-increasing*, genes significantly upregulated between days 20 and 30 ($20–30_{up}^{WT}$) and also days 30 and 60 ($30–60_{up}^{WT}$); *early-stable* genes, present in $20–30_{up}^{WT}$ and $20–60_{up}^{WT}$ but not $30–60_{up}^{WT}$; *early-transient* ($20–30_{up}^{WT}$ but not $20–60_{up}^{WT}$); and *late* ($30–60_{up}^{WT}$ but not $20–30_{up}^{WT}$). **c** SZ GWAS enrichment in each transcriptional program, further split into genes that are down-regulated in $DLG2^{-/-}$ lines at day 30 (e.g., early-stable$^{-/-}$) and those that are not (e.g., early-stable$^{WT only}$). One-sided tests were performed using MAGMA, conditioning on all expressed genes ($all^{WT+KO}$); raw and Bonferroni-corrected $p$ values are given ($P^{GWAS}$, $P^{GWAS}_{corrected}$), where correction is for the 7 gene-sets tested. **d** A one-sided, Fisher's Exact Test was used to identify programs over-represented for the targets of key regulators when compared to $all^{WT+KO}$; both raw and Bonferroni-corrected $p$ values are given ($P^{overlap}$, $P^{overlap}_{corrected}$), where correction is for the 21 tests performed (3 programs x 7 regulators). All 10 program-regulator enrichments with corrected $P < 0.05$ were taken forward for genetic analysis. Two-sided tests were performed in MAGMA to investigate whether regulator targets were more highly enriched for SZ association than other genes in that program, conditioning on $all^{WT+KO}$ and the program as a whole. Raw and Bonferroni-corrected $p$ values are given ($P^{GWAS}$, $P^{GWAS}_{corrected}$), where correction is for the 10 sets tested. **e** SZ GWAS enrichment in GO terms with independent evidence of over-representation amongst early-increasing$^{-/-}$ genes, two-sided tests were performed using MAGMA, conditioning on all expressed and all early-increasing$^{-/-}$ genes. Raw and Bonferroni-corrected $p$ values are given ($P^{GWAS}$, $P^{GWAS}_{corrected}$), where correction is for the 16 terms tested. Bold indicates tests surviving Bonferroni correction. Source data are provided as a Source Data file.

LoFi genes[12]. We predicted that LoFi genes would primarily be concentrated in earlier transcriptional programs where the impact of disruption is potentially more severe. LoFi genes were over-represented in all early neurogenic programs but notably depleted in the late set (Fig. 6a). LoFi SZ association was captured by the overlap with early-stable$^{-/-}$ and early-increasing$^{-/-}$, localizing the GWAS signal to a fraction of LoFi genes (less than a third) located in specific neurogenic pathways (Fig. 6b).

Under our proposed model, early-transient$^{-/-}$ initiates activation of other early neurogenic programs, thus its dysregulation has the potential to cause more profound developmental deficits. We speculated that – while displaying no evidence for SZ GWAS association – LoFi genes in early-transient$^{-/-}$ would be enriched for rare mutations linked to SZ and/or more severe neurodevelopmental disorders. All early neurogenic programs displayed a markedly elevated rate of *de novo* LoF mutations relative to $all^{WT+KO}$ that was captured by LoFi genes: early-transient$^{-/-}$ for mutations identified in NDD and ASD cases[47]; early-stable$^{-/-}$ for NDD, ASD and SZ[16]; and early-increasing$^{-/-}$ for NDD (Fig. 6c). *De novo* LoF mutations from unaffected siblings of ASD cases[47] showed no elevation. In all three programs, a clear gradient of effect was evident from NDD (largest elevation in rate) to ASD to SZ, visible only in LoFi genes (Fig. 6d). A modest gradient was also evident for LoFi genes lying outside early neurogenic programs ('Other LoFi genes', Fig. 6d), despite *de novo* rates not being robustly elevated here. This suggests the existence of additional biological pathways harboring disease-associated LoFi genes.

Given the robust rare variant enrichment across multiple disorders, we investigated whether neurogenic programs are also enriched for common variants contributing to disorders other than SZ, analyzing a range of conditions with which SZ is known to share heritability: ASD[48]; attention-deficit/hyperactivity disorder (ADHD)[49]; bipolar disorder (BP)[50]; and major depressive disorder (MDD)[51]. Since altered cognitive function is a feature of all these disorders, we also tested enrichment for common variants linked to IQ[52]. All disorders showed evidence for common variant enrichment in one or more early neurogenic program that was again captured by LoFi genes (Fig. 6e, f). In contrast, common variants conferring risk for the neurodegenerative disorder Alzheimer's disease (AD)[53] were not enriched. Whereas rare variant enrichment was concentrated towards the initial stages of the transcriptional cascade, GWAS association was confined to later stages

(Fig. 6c–f). Dysregulation of transcriptional programs underlying cortical excitatory neurogenesis thus contributes to a wide spectrum of neuropsychiatric disorders. Furthermore, robust enrichment of early-stable$^{-/-}$ and early-increasing$^{-/-}$ for IQ association (Fig. 6e, f) suggests that perturbation of neurogenic programs may contribute to the emergence of cognitive symptoms in these disorders.

**SZ association seen in mature neurons captured by neurogenic programs.** DLG2 plays an essential role[26] in scaffolding mature postsynaptic complexes implicated in SZ[4,5,8–11]. Our data indicates that it also regulates early developmental pathways harboring SZ genetic risk. To further explore the relationship between developmental and adult disease mechanisms, we investigated the extent to which genes from neurogenic programs contribute to SZ-associated biology in mature excitatory neurons. GWAS association has previously been noted in genes with relatively high expression in CA1 pyramidal neurons compared to other brain cell-types[13,54]. Although different to the sub-types generated by our in vitro protocol, we reasoned that developmental processes shared between these two dorsal forebrain-derived neuronal types are likely to account for a substantial proportion of the neurogenic programs we have identified. Taking the 10% of genes with the highest CA1 pyramidal neuron specificity score[54] (pyramidal$^{high}$) we investigated their overlap with neurogenic programs. Pyramidal$^{high}$ genes were over-represented in early-stable$^{-/-}$, early-increasing$^{-/-}$ and late sets (Fig. 7a). This overlap captured GWAS association in pyramidal$^{high}$, but not early-stable$^{-/-}$ or early-increasing$^{-/-}$ (Fig. 7b). In contrast to the late program as a whole (Figs. 4c and 6a), genes in the late-pyramidal$^{high}$ overlap were enriched for SZ association (Fig. 7b) and LoFi genes ($OR = 1.43$, $P = 0.035$). Late-pyramidal$^{high}$ genes also displayed a pattern of enrichment for regulatory targets almost identical to that of early-increasing$^{-/-}$ (Figs. 4d and 7c), linking this subset into the terminal phase of the hypothesized transcriptional cascade.

These analyses suggest that the SZ association seen in pyramidal excitatory neurons[13,54] may arise from molecular pathways contributing to early neurogenesis that remain active in post-natal life. To investigate the nature of these pathways, we performed a functional analysis of SZ-associated gene-sets from Fig. 7b. Pyramidal$^{high}$ genes overlapping early-stable$^{-/-}$, early-increasing$^{-/-}$ and late sets were over-represented for GO

terms linked to dendrite/spine development, calcium-mediated exocytosis, postsynaptic signaling and synaptic plasticity (Supplementary Data 9). In contrast, genes unique to early neurogenic programs were over-represented for terms linked to the regulation of transcription/neurogenesis, axonogenesis, axon guidance, pre-synaptic function and sodium channel

activity (Supplementary Data 9). Thus it appears to be primarily postsynaptic processes regulating the formation, function and plasticity of synaptic connections throughout development (from the pre-natal period into adulthood) that underlies the mature neuronal contribution to SZ encapsulated by pyramidal[high].

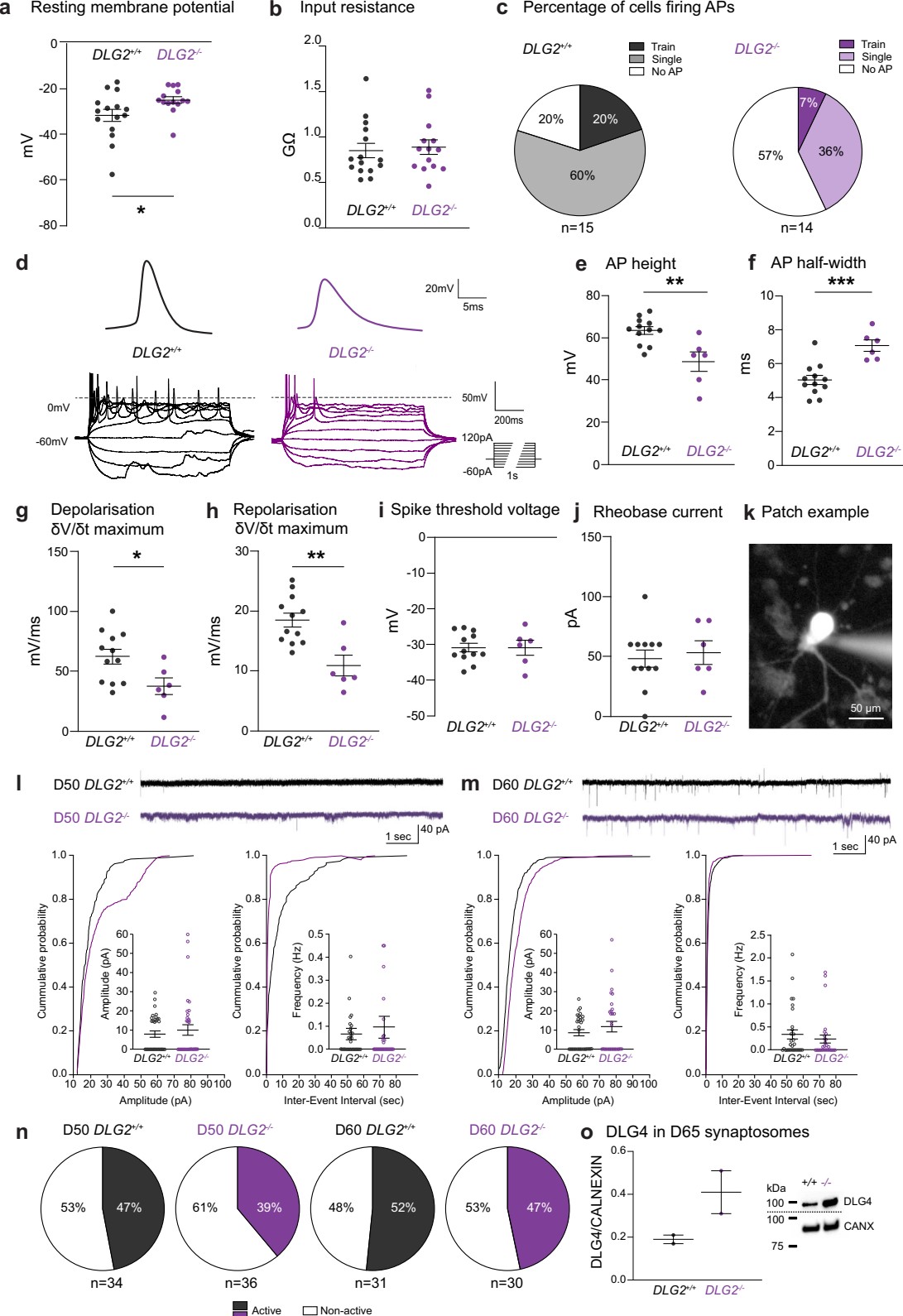

**Fig. 5 Electrophysiological properties of $DLG2^{-/-}$ neurons. a** Resting membrane potential ($t_{27} = 2.151$, $P = 0.0406$) and **b** input resistance ($t_{27} = 0.3366$, $P = 0.7390$) of day 50 WT and $DLG2^{-/-}$ neurons ($n = 15$ and 14, respectively, unpaired two-tailed $t$ test). **c** Percentage of cells firing action potentials (APs) upon current step injection. **d** Example traces of first overshooting AP and APs evoked by current step injection ($-60$ pA to 120 pA, increment 20 pA, duration 1 s). **e** AP height ($t_{16} = 3.661$, $P = 0.0021$), **f** AP half-width ($t_{16} = 4.462$, $P = 0.0004$), **g** AP maximum depolarizing speed ($t_{16} = 2.463$, $P = 0.0255$), **h** AP maximum repolarising speed ($t_{16} = 3.728$, $P = 0.0018$), **i** spike threshold voltage ($t_{16} = 0.004093$, $P = 0.9968$) and **j** rheobase current ($t_{16} = 0.4061$, $P = 0.6900$) of day 50 WT and $DLG2^{-/-}$ neurons ($n = 12$ and 6, respectively, unpaired two-tailed $t$ test) are shown. **k** Example of day 50 neuron being whole-cell patch clamped with fluorescent dye injection. Spontaneous excitatory postsynaptic current (sEPSC) examples from day 50 (**l**) and 60 (**m**) neurons. Both the amplitude and frequency of day 50 and 60 neurons from WT and $DLG2^{-/-}$ neurons were comparable by unpaired two-tailed $t$ test (day 50, amplitude: $t_{68} = 0.6974$, $P = 0.4879$, $n = 34$ and 36 for WT and KO; frequency: $t_{66} = 0.5467$, $P = 0.5865$, $n = 33$ and 35 for WT and KO; day 60 amplitude: $t_{59} = 1.021$, $P = 0.3114$, $n = 31$ and 30 for WT and KO; frequency: $t_{58} = 0.7671$, $P = 0.4464$, $n = 30$ each for WT and KO). **n** Percentage of cells displaying sEPSCs. **o** Western blot analysis of DLG4 in synaptosomes of day 65 WT and $DLG2^{-/-}$ neurons, displaying trend towards increased DLG4 expression in $DLG2^{-/-}$ neurons ($t_2 = 2.157$, $P = 0.1637$, $n = 2$, unpaired two-tailed $t$ test). *$p < 0.05$; **$p < 0.01$; ***$p < 0.001$. All data presented as mean $\pm$ SEM and are from two independent experiments. Source data are provided as a Source Data file.

**Neurogenic programs identified in vitro are expressed in human fetal cortex.** Early neurogenic programs are enriched for variants contributing to cognitive function and the pathogenesis of neuropsychiatric disorders. However, these programs were identified from bulk RNAseq data in vitro and it remains to be shown that their constituent genes are actively co-expressed in the appropriate cell-types during cortical excitatory neurogenesis in vivo. To address this, we extracted gene expression data for cell-types spanning cortical excitatory neurodevelopment from a single-cell RNAseq study of human fetal brain[32]. After normalizing the expression for each gene across all cells, we calculated the average expression for each gene in each cell-type/stage of development available: early radial glia (early RG), RG, intermediate progenitor cells (IPCs), transitioning cells (intermediate between progenitors and neurons), newborn and developing neurons (Methods). We then plotted the expression of each program (mean and standard error of gene-level averages) in each cell-type/stage and tested for differences in expression between successive types/stages (Fig. 7d, Supplementary Data 10). The expression profile seen for each program in vitro was recapitulated across neurodevelopmental cell-types from human fetal cortex (Fig. 7d). Notably, while other programs in the cascade were significantly upregulated during the transition from progenitors to neurons, early-transient$^{-/-}$ expression was found to be low in early RG, rising in more mature neural progenitor cells (NPCs) then declining in neurons. This is consistent with its predicted role in shaping neuronal sub-type identity, which recent evidence indicates is determined by the internal state of NPCs immediately prior to their exit from the cell-cycle[55].

## Discussion

A complex choreography of cell proliferation, specification, growth, migration and network formation underlies brain development. To date, limited progress has been made pinpointing aspects of this process disrupted in neuropsychiatric disorders. Here we uncover distinct gene expression programs expressed during early excitatory corticoneurogenesis in vitro and in human fetal cortex (Fig. 7d). These programs are enriched for variants contributing to a wide spectrum of disorders and cognitive function (Fig. 6). The consistency of these enrichments is noteworthy, with multiple associations identified for each early neurogenic program. These programs harbor well-supported risk genes for complex and Mendelian disorders, some of which are highlighted in Fig. 8a. This convergence of genetic evidence suggests that these programs play an etiological role in a wide range of psychiatric disorders.

Each program has a unique gene expression profile and molecular composition, indicating a distinct functional role during neurogenesis. Based on our findings we propose that they form a transcriptional cascade regulating neuronal growth, migration, differentiation and network formation (Fig. 8a). Computational analyses of gene/mRNA regulatory interactions implicate known neurodevelopmental disorder risk genes (CHD8, TCF4, FMRP, BCL11B and TBR1) as regulators of this cascade and reveal pathways through which they may contribute to disease (Fig. 4d). Supporting this model, down-regulation of neurogenic programs in $DLG2^{-/-}$ lines is accompanied by deficits that match their predicted function: impaired migration; simplified neuronal morphology; immature action potential generation; and delayed expression of cell-type identity (reduced expression of CTIP2 protein (Fig. 2e, h, i) and TBR1 mRNA (Supplementary Data 3)). Interestingly, voltage-gated sodium and L-type calcium channels present in early neurogenic programs are not only involved in the generation and control of action potentials[56], but are also known to impact neuronal growth and migration[57,58]. Further experimental work is required to more precisely delineate phenotypes associated with the disruption of individual programs and the risk genes they harbor, and to map out regulatory interactions shaping their expression and activity, testing predictions (Fig. 8a). Here we focus on phenotypes expressed by individual newborn excitatory neurons; in future studies it will be important to investigate the persistence of these phenotypes and explore longer-term effects on neuronal circuit formation and function.

A clear pattern of enrichment was evident across early neurogenic programs (Fig. 6c–f). Rare damaging mutations contributing to more severe disorders were concentrated in initial stages of the cascade, impacting both progenitors (early-transient$^{-/-}$, Fig. 7d) and neurons. Common variant association was restricted to neuronally expressed pathways (early-stable$^{-/-}$, early-increasing$^{-/-}$). It has been proposed that adult and childhood disorders lie on an etiological and neurodevelopmental continuum, the more severe the disorder the greater the contribution from rare, damaging mutations and the earlier their developmental impact[59–61] (Fig. 8b). Our data support this model and ground it in developmental neurobiology, embedding genetic risk for multiple disorders in a common pathophysiological framework.

Genetic risk for all disorders was concentrated in LoFi genes, indicating wider relevance for these genes than previously appreciated and providing insight into their pathophysiological roles. Being under high selective constraint, LoFi genes profoundly impact development through to sexual maturity. It has not been clear whether LoFi genes harboring pathogenic mutations are distributed across diverse pathways shaping pre-/postnatal growth or are concentrated in specific pathways and/or stages of development. Our analyses reveal that not all neurodevelopmental pathways are enriched for LoFi genes (Fig. 6a), and that the subset of LoFi genes (~40%) concentrated in early

neurogenic programs capture virtually all common and rare variant LoFi association across a wide spectrum of disorders (Fig. 6). While early-transient$^{-/-}$ expression is limited to initial stages of neurogenesis (peaking as RG mature, Fig. 7d), other programs are upregulated during the NPC-neuron transition and persist as neurons develop (Fig. 7d), shaping their morphology, function and possibly connectivity (Fig. 8a).

SZ GWAS association previously noted in CA1 pyramidal neurons[13,54] was captured by a subset of neurogenic genes contributing to dendrite/spine/synapse formation, signaling and plasticity. These processes underlie the establishment and maturation of neuronal circuits[62] and their learning-dependent modification in adults[63]. The most parsimonious explanation is that SZ common variants act largely via disruption of early brain

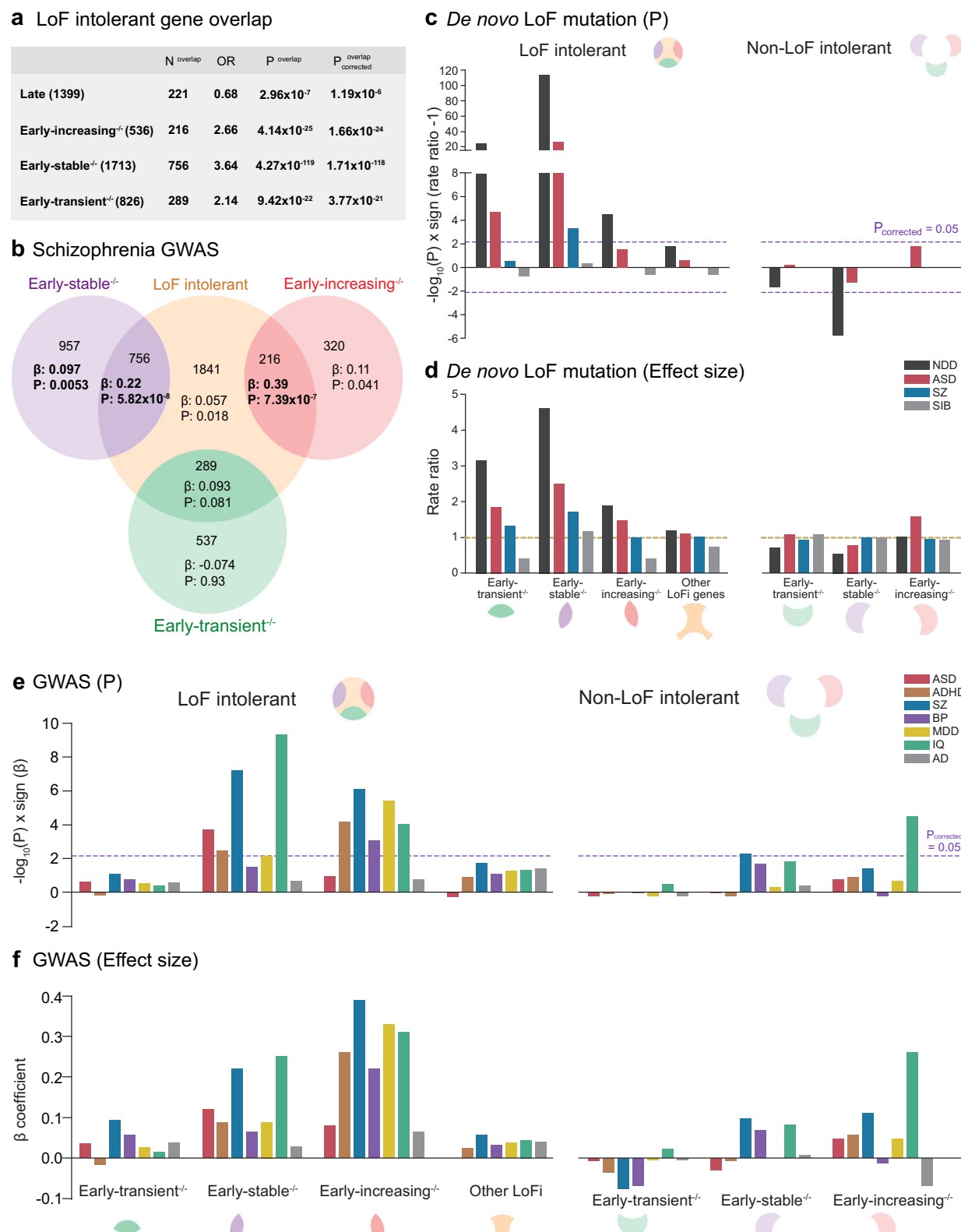

**a** LoF intolerant gene overlap

| | N overlap | OR | P overlap | P overlap corrected |
|---|---|---|---|---|
| **Late (1399)** | **221** | **0.68** | **2.96x10$^{-7}$** | **1.19x10$^{-6}$** |
| **Early-increasing$^{-/-}$ (536)** | **216** | **2.66** | **4.14x10$^{-25}$** | **1.66x10$^{-24}$** |
| **Early-stable$^{-/-}$ (1713)** | **756** | **3.64** | **4.27x10$^{-119}$** | **1.71x10$^{-118}$** |
| **Early-transient$^{-/-}$ (826)** | **289** | **2.14** | **9.42x10$^{-22}$** | **3.77x10$^{-21}$** |

**b** Schizophrenia GWAS

**c** *De novo* LoF mutation (P)

**d** *De novo* LoF mutation (Effect size)

**e** GWAS (P)

**f** GWAS (Effect size)

**Fig. 6 Neuropsychiatric disorder risk variants localize to LoF intolerant genes in early neurogenic transcriptional programs. a** Identification of programs enriched/depleted for LoFi genes ($P^{overlap}$) relative to all expressed genes (two-sided Fisher's Exact Test). Raw and Bonferroni-corrected p values given ($P^{overlap}$, $P^{overlap}_{corrected}$), where correction is for the 4 tests performed. **b** LoFi genes were partitioned based on their overlap with early neurogenic programs. Each segment of the Venn diagram shows the number of genes in each subset and the regression coefficient (β) and (uncorrected) p value (P) for schizophrenia (SZ) common variant enrichment (one-sided MAGMA test), conditioning on $all^{WT+KO}$. Bold indicates enrichments surviving Bonferroni correction for 7 tests. **c** A two-sided Poisson rate ratio test was used to identify programs (partitioned by LoFi gene overlap) enriched for *de novo* LoF mutations from individuals with intellectual disability/severe neurodevelopmental delay (ID/NDD), autism spectrum disorder (ASD) and SZ when compared to all other expressed genes. Unaffected siblings (SIB) of ASD cases were analyzed as a control. Data points lying above the x axis indicate an increased rate (rate ratio > 1), those below indicate a reduced rate (rate ratio < 1). Dotted lines indicate $P_{corrected} = 0.05$ following Bonferroni correction for 7 tests (4 LoFi + 3 non-LoFi sets tested for each disorder). **d** Rate ratios (genes in set versus all other expressed genes) from tests shown in **c**. Dotted line shows rate ratio of 1 (i.e., rate of mutations in set equals that in all other genes). **e** Programs (partitioned by LoFi gene overlap) were tested for enrichment (one-sided MAGMA test) in common variants contributing to ASD, attention-deficit/hyperactivity disorder (ADHD), bipolar disorder (BP), major depressive disorder (MDD) and Alzheimer's disease (AD). Values for SZ (Fig. 6b) included for comparison. Dotted lines indicate $P_{corrected} = 0.05$ following Bonferroni correction for 7 tests (4 LoFi + 3 non-LoFi sets tested for each disorder). **f** Effect sizes (β coefficient from MAGMA gene-set enrichment test) for tests shown in **e**. Bold indicates tests surviving Bonferroni correction. Source data are provided as a Source Data file.

development, as neurogenic programs harbor GWAS association beyond their overlap with pyramidal[high] (Fig. 7b); and SZ shares extensive SNP heritability with early-onset disorders[48,49]. However, effect sizes are greater for shared genes (Fig. 7b) – although not significantly so – and SZ onset extends from late childhood well into adulthood[64]. We hypothesize that vulnerability to SZ is primarily established during early neurodevelopment, and that this is subsequently compounded by a gradual accumulation of deficits during circuit maturation due to both external stressors and the impaired function of neurogenic pathways that remain operant throughout childhood and into adulthood.

While *DLG2* knockout led to the identification of disease-associated programs and allowed us to investigate cellular phenotypes associated with their dysregulation, *DLG2* itself has yet to reach the status of a canonical SZ/ASD risk gene. DLG2 is primarily known for its role as a postsynaptic scaffold protein in mature neurons, where it is required for normal formation of NMDA receptor signaling complexes[26]. We show that *DLG2* expression is also important for cortical excitatory neurodevelopment, but the mechanism by which it operates remains to be determined. Based on its known function and the involvement of invertebrate *Dlg* in the developmental *Scrib* signaling module[25], DLG2 may link cell-surface receptors to signal transduction pathways regulating the activation of neurogenic programs (Supplementary Fig. 9). We hypothesize that stochastic signaling in $DLG2^{-/-}$ lines due to impaired complex formation could delay and impair transcriptional activation, disrupting the orchestration of events required for normal development and the specification of neuronal properties. Precise timing is crucial during brain development, where the correct dendritic morphology, axonal length and electrical properties are required for normal circuit formation and function. Consequently, even transient perturbation of neurogenesis may have a profound impact on fine-grained neuronal wiring, network activity and ultimately perception, cognition and behavior.

Although much remains to be uncovered, our findings sketch the foundations for an integrated etiological model of psychiatric genetic disorders and their developmental origins.

## Methods

**hESC culture.** H7 hESC line (WA07) was purchased from Wi Cell, USA. All hESC work were performed in accordance with Cardiff University's regulation under Health and Safety Executive approval (GM130/14.3) and WiCell's MTA and SLA. All hESC lines were maintained at 37 °C and 5% CO$_2$ in 6 well cell culture plates (Greiner) coated with 1% Matrigel hESC-Qualified Matrix (Corning) prepared in Dulbecco's Modified Eagle Medium: Nutrient Mixture F-12 (DMEM/F12, Thermo Fisher Scientific). Cells were fed daily with Essential 8 medium (E8, Thermo Fisher Scientific) and passaged at 80% confluency using Versene solution (Thermo Fisher Scientific) for 1.5 min at 37 °C followed by manual dissociation with a serological

pipette. All cells were kept below passage 25 and confirmed as negative for mycoplasma infection.

**DLG2 Knockout hESC line generation.** Two guide RNAs targeting exon 22 (Supplementary Fig. 1) of the human *DLG2* gene, covering the first PDZ domain, were designed using a web-based tool (crispr.mit.edu) and cloned into two plasmids containing D10A nickase mutant Cas9 with GFP (PX461) or Puromycin resistant gene (PX462)[65]. pSpCas9n(BB)-2A-GFP (PX461) and pSpCas9n(BB)-2A-Puro (PX462) was a gift from Feng Zhang (For PX461, Addgene plasmid#48140; http://n2t.net/addgene:48140; RRID:Addgene_48140; For PX462, Addgene plasmid #48141; http://n2t.net/addgene:48141; RRID:Addgene_48141). H7 hESCs (WiCell) were nucleofected using P4 solution and CB150 programme (Lonza) with 5 µg of plasmids, FACS sorted on the following day and plated at a low density (~70 cells/cm$^2$) for clonal isolation. 19 clonal populations were established with 6 WT and 13 mutant lines after targeted sequencing of the exon 22. One WT and two homozygous knockout lines were chosen for study: our WT and KO lines therefore originate from the same H7 parental line and have gone through the same process of nucleofection and FACS sorting together.

**Genetic validation.** The gRNA pair had zero predicted off-target nickase sites (Supplementary Fig. 2). Even though we did not use a wild-type Cas9 nuclease (where only a single gRNA is required to create a double-stranded break), we further checked genic predicted off-target sites for each individual gRNA by PCR and Sanger sequencing (GATC & LGC). Out of 30 sites identified, we randomly selected 14 (7 for each gRNA) for validation. No mutations relative to WT were present at any site (Supplementary Data 1). In addition, genotyping on the Illumina PsychArray v1.1 revealed no CNV insertions/deletions in either $DLG2^{-/-}$ line relative to WT (Supplementary Fig. 5).

**Cortical differentiation.** Differentiation to cortical projection neurons (Fig. 1b) was achieved using the dual SMAD inhibition protocol[28] with modifications (embryoid body to monolayer and replacement of KSR medium with N2B27 medium) suggested by Cambray et al.[29]. Prior to differentiation Versene treatment and mechanical dissociation was used to passage hESCs at ~100,000 cells per well into 12 well cell culture plates (Greiner) coated with 1% Matrigel Growth Factor Reduced (GFR) Basement Membrane matrix (Corning) in DMEM/F12, cells were maintained in E8 medium at 37 °C and 5% CO$_2$ until 90% confluent. At day 0 of the differentiation E8 media was replaced with N2B27-RA neuronal differentiation media consisting of: 2/3 DMEM/F12, 1/3 Neurobasal (Thermo Fisher Scientific), 1x N-2 (Thermo Fisher Scientific), 1x B27 Supplement minus vitamin A (Thermo Fisher Scientific), 1x Pen Step Glutamine (Thermo Fisher Scientific) and 50 µM 2-Mercaptoethanol (Thermo Fisher Scientific), which was supplemented with 100 nM LDN193189 (Cambridge Biosciences) and 10 µM SB431542 (Stratech Scientific) for the first 10 days only (the neural induction period). At day 10 cells were passaged at a 2:3 ratio into 12 well cell culture plates coated with 15 µg/ml human plasma fibronectin (Merck) in Dulbecco's phosphate-buffered saline (DPBS, Thermo Fisher Scientific), passage was as previously described with the addition of a 1 h incubation with 10 µM Y27632 Dihydrochloride (ROCK inhibitor, Stratech Scientific) prior to Versene dissociation. During days 10–20 of differentiation cells were maintained in N2B27-RA (without LDN193189 or SB431542 supplementation) and at day 20 in a 1:4 ratio into 24 well cell culture plates (Greiner) sequentially coated with 10 µg/ml poly-d-lysine hydrobromide (PDL, Sigma) and 15 µg/ml laminin (Sigma) in DPBS. Vitamin A was added to the differentiation media at day 26, standard 1x B27 Supplement (Thermo Fisher Scientific) replacing 1x B27 Supplement minus vitamin A, and cells were maintained in the resulting N2B27 + RA media for the remainder of the differentiation. Cells maintained to day 40 received no additional passage beyond

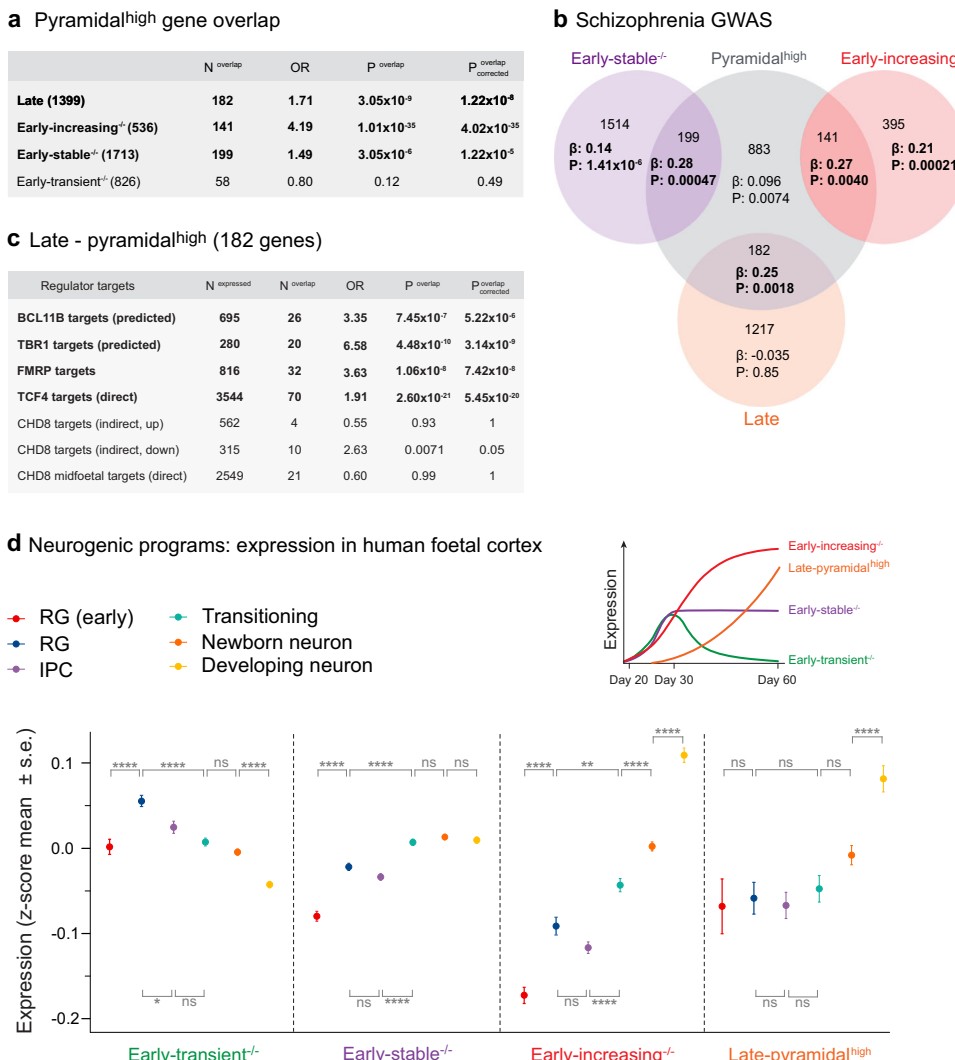

**a** Pyramidal^high^ gene overlap

| | N overlap | OR | P overlap | P overlap corrected |
|---|---|---|---|---|
| **Late (1399)** | **182** | **1.71** | **3.05x10⁻⁹** | **1.22x10⁻⁸** |
| **Early-increasing⁻/⁻ (536)** | **141** | **4.19** | **1.01x10⁻³⁵** | **4.02x10⁻³⁵** |
| **Early-stable⁻/⁻ (1713)** | **199** | **1.49** | **3.05x10⁻⁶** | **1.22x10⁻⁵** |
| Early-transient⁻/⁻ (826) | 58 | 0.80 | 0.12 | 0.49 |

**b** Schizophrenia GWAS

**c** Late - pyramidal^high^ (182 genes)

| Regulator targets | N expressed | N overlap | OR | P overlap | P overlap corrected |
|---|---|---|---|---|---|
| **BCL11B targets (predicted)** | **695** | **26** | **3.35** | **7.45x10⁻⁷** | **5.22x10⁻⁶** |
| **TBR1 targets (predicted)** | **280** | **20** | **6.58** | **4.48x10⁻¹⁰** | **3.14x10⁻⁹** |
| **FMRP targets** | **816** | **32** | **3.63** | **1.06x10⁻⁸** | **7.42x10⁻⁸** |
| **TCF4 targets (direct)** | **3544** | **70** | **1.91** | **2.60x10⁻²¹** | **5.45x10⁻²⁰** |
| CHD8 targets (indirect, up) | 562 | 4 | 0.55 | 0.93 | 1 |
| CHD8 targets (indirect, down) | 315 | 10 | 2.63 | 0.0071 | 0.05 |
| CHD8 midfoetal targets (direct) | 2549 | 21 | 0.60 | 0.99 | 1 |

**d** Neurogenic programs: expression in human foetal cortex

**Fig. 7 Expression of early neurogenic programs in neurodevelopmental cell-types from human fetal cortex. a** Identification of programs enriched/depleted for genes with high expression in CA1 pyramidal neurons relative to other brain cell-types (pyramidal^high^); enrichment compared to all expressed genes (two-sided Fisher's Exact Test). Raw and Bonferroni-corrected $p$ values given ($P$overlap, $P$overlap_corrected), correction is for the 4 tests performed. **b** Pyramidal^high^ genes were partitioned based on their overlap with neurogenic programs. Each segment of the Venn diagram shows the number of genes in each subset and the regression coefficient ($\beta$) and uncorrected $p$ value ($P$) for SZ common variant enrichment (one-sided MAGMA test), conditioning on all expressed genes. Bold indicates enrichments surviving correction for 7 tests. **c** A one-sided Fisher's Exact Test was used to identify key regulators whose known/predicted targets are over-represented amongst pyramidal^high^ genes overlapping the late neurogenic program when compared to all expressed genes; both raw and Bonferroni-corrected $p$ values are given ($P$overlap, $P$overlap_corrected), where correction is for the 7 regulator sets tested. **d** Cells corresponding to distinct neurodevelopmental cell-types (RG radial glia; IPC Intermediate progenitor cell), including cell-types at different stages of maturity, were identified and extracted from a previously published single-cell RNA-seq study of human fetal cortex across peak stages of neurogenesis[32] (Methods). >80% of genes belonging to each in vitro-defined early neurogenic program and 55% of the late-pyramidal^high^ set (top RHS) were present in the fetal data (early-transient⁻/⁻ $n = 665$ genes; early-stable⁻/⁻ $n = 1484$ genes; early-increasing⁻/⁻ $n = 431$ genes; late-pyramidal^high^ $n = 101$ genes). For each program, the mean and standard error of gene-level expression averages (see main text/Methods) were calculated for each fetal cell-type/developmental stage. The fetal data captures both direct (RG ➔ neuron) and indirect (RG ➔ IPC ➔ neuron) neurogenic pathways. The (deep layer) neurons present in our day 30–60 cultures in vitro are predominantly born via the direct neurogenic pathway. For each program, differences between gene-level averages for successive cell-types/stages were evaluated using a two-tailed Student's $t$ test and $p$ values Bonferroni-corrected for 6 pairwise comparisons. *$p < 0.05$; **$p < 0.01$; ****$p < 0.0001$. All data presented as mean ± SEM. Bold indicates tests surviving Bonferroni correction. The exact $p$ values are provided in Supplementary Data 10.

passage 2 at day 20 while cells kept beyond day 40 received a third passage at day 30, 1:2 onto PDL-laminin as previously described. In all cases cells maintained past day 30 were fed with N2B27 + RA supplemented with 2 µg/ml laminin once weekly to prevent cell detachment from the culture plates.

**Immunocytochemistry**. Cells were fixed in 4% paraformaldehyde (PFA, Sigma) in PBS for 20 min at 4 °C followed by a 1 h room temperature incubation in blocking solution of 5% donkey serum (Biosera) in 0.3% Triton-X-100 (Sigma) in PBS (0.3% PBST). Primary antibodies, used at an assay dependent concentration (see 'Antibody concentration'), were diluted in blocking solution and incubated with cells overnight at 4 °C. Following removal of primary antibody solution and 3 PBS washes, cells were incubated in the dark for 2 h at room temperature with appropriate Alexa Fluor secondary antibodies (Thermo Fisher Scientific) diluted 1:500 with blocking solution. After an additional 2 PBS washes cells were counterstained with DAPI nucleic acid stain (Thermo Fisher Scientific), diluted 1:1000 with PBS, for 5 mins at room

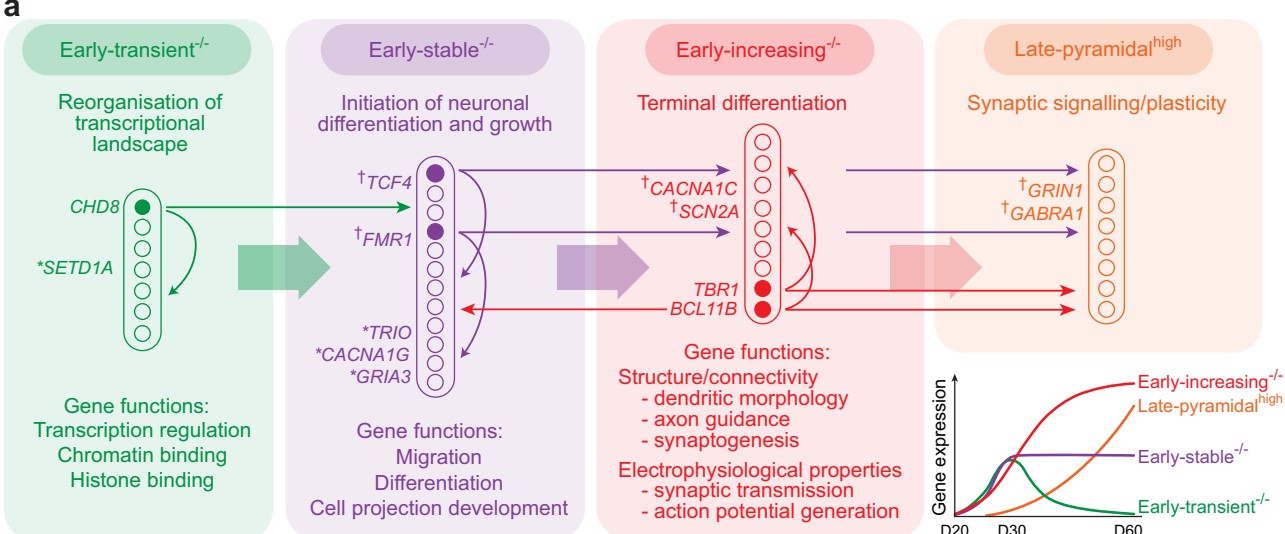

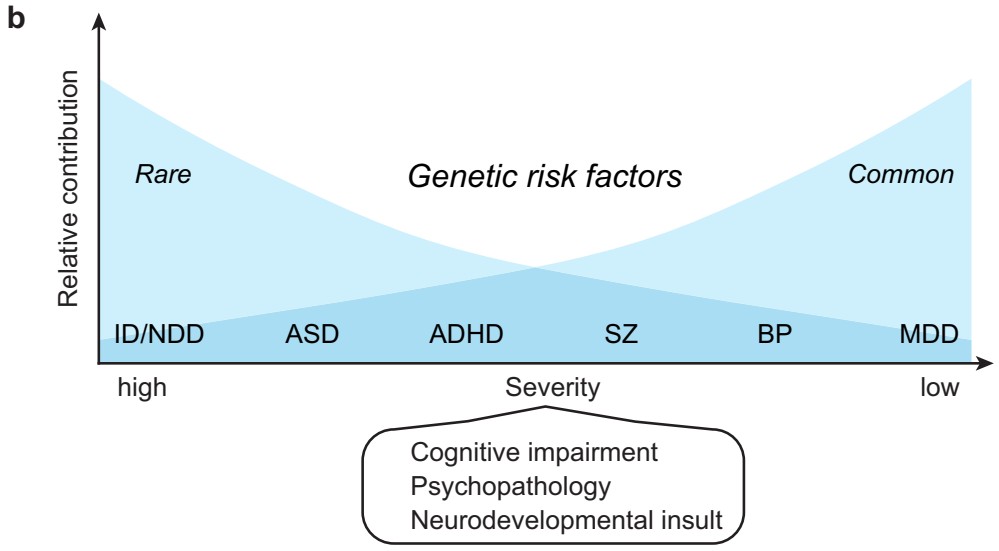

**Fig. 8 Model of disease pathophysiology in early corticoneurogenesis. a** Summary of main GO term enrichments for each disease-associated transcriptional program (Supplementary Data 7) and significant regulatory interactions between them. Key regulators, genome-wide significant rare coding variants from[84] and rare mutations causing Mendelian neurodevelopmental syndromes are denoted. **b** Neurodevelopmental continuum/gradient model. Disorders shown are: ID/NDD intellectual disability/severe neurodevelopmental delay; ASD autism spectrum disorders; ADHD attention-deficit/hyperactivity disorder; SZ schizophrenia; BP bipolar disorder; MDD major depressive disorder.

temperature and following a final PBS wash, mounted using Dako Fluorescence Mounting Medium (Agilent) and glass coverslips. Imaging was with either the LSM710 confocal microscope (Zeiss) using Zen 2012 SP2 (black) v11.02.190 (Zeiss), LAS X for DMI6000B inverted microscope (Leica) or Cellinsight Cx7 High-Content Screening Platform (Thermo Fisher Scientific) with HCS Studio Cell Analysis software v6.6.0 (Thermo Fisher Scientific) used for quantification.

**Western blotting**. Total protein was extracted from dissociated cultured cells by incubating in 1x RIPA buffer (New England Biolabs) with added MS-SAFE Protease and Phosphatase Inhibitor (Sigma) for 30 min on ice with regular vortexing, concentration was determined using a DC Protein Assay (BioRad) quantified with the CLARIOstar microplate reader (BMG Labtech). Proteins for western blotting were incubated with Bolt LDS sample buffer (Thermo Fisher Scientific) and Bolt Sample Reducing Agent (Thermo Fisher Scientific) for 10 min at 70 °C before loading into Bolt 4–12% Bis-Tris Plus gels (Thermo

Fisher Scientific). Gels were run at 120 V for 2–3 h in Bolt MES SDS Running Buffer (Thermo Fisher Scientific) prior to protein transfer to Amersham Protran nitrocellulose blotting membrane (GE Healthcare) using a Mini Trans-Blot Cell (BioRad) and Bolt Transfer Buffer (Thermo Fisher Scientific) run at 120 V for 1 h 45 min. Transfer was confirmed by visualizing protein bands with 0.1% Ponceau S (Sigma) in 5% acetic acid (Sigma) followed by repeated $H_2O$ washes to remove the stain.

Following transfer, membranes were incubated in a blocking solution of 5% milk in TBST, 0.1% TWEEN 20 (Sigma) in TBS (Formedium), for 1 h at room temperature. Primary antibodies, used at an assay dependent concentration, were diluted with blocking solution prior to incubation with membranes overnight at 4 °C. Following 3 TBST washes, membranes were incubated in the dark for 1 h at room temperature with IRDye secondary antibodies (LI-COR) diluted 1:15000 with blocking solution. After 3 TBS washes staining was visualized using the Odyssey CLx Imaging System (LI-COR) and Image Studio Lite Version 5.2 (LI-COR).

**Antibody concentration.**

| Antibody | Company | Identifier catalog no. | Dilution in ICC | Dilution in WB |
|---|---|---|---|---|
| Rabbit polyclonal anti-Calnexin | Abcam | ab22595 | N/A | 1:5000 |
| Rat monoclonal anti-CTIP2 [25B6] | Abcam | ab18465 | 1:500 | N/A |
| Rabbit polyclonal anti-CTIP2 | Abcam | ab70453 | N/A | 1:1000 |
| Rabbit monoclonal anti-DARPP32 [EP721Y] | AbCam | ab40802 | 1:500 | N/A |
| Mouse polyclonal anti-DLX1 | AbCam | ab167575 | 1:100 | N/A |
| Rabbit polyclonal anti-FOXG1 | Abcam | ab18259 | 1:250 | N/A |
| Mouse monoclonal anti-FOXP1 [JC12] | Abcam | ab32010 | 1:800 | N/A |
| Rabbit polyclonal anti-GABA | Sigma | A2052 | 1:500 | N/A |
| Rabbit polyclonal anti-GAPDH | Abcam | ab9485 | N/A | 1:5000 |
| Goat polyclonal anti-GBX2 | Antibodies.com | A84236 | 1:200 | N/A |
| Mouse monoclonal anti-KI67 [B56] | BD Pharmingen | 550609 | 1:150 | 1:1000 |
| Mouse monoclonal anti-MAP2 [AP-20] | Sigma-Aldrich | M1406 | 1:250 | N/A |
| Rabbit polyclonal anti-MAP2 | Merck Millipore | AB5622 | N/A | 1:1000 |
| Rabbit monoclonal anti-NANOG [D73G4] | Cell Signaling Technology | 4903 | 1:400 | N/A |
| Rabbit polyclonal anti-NEUN | Merck Millipore | ABN78 | 1:500 | 1:500 |
| Rabbit monoclonal anti-NKX2.1 [EP1584Y] | Abcam | ab76013 | 1:1000 | N/A |
| Rabbit monoclonal anti-OCT4 [C30A3] | Cell Signaling Technology | 2840 | 1:400 | N/A |
| Mouse monoclonal anti-OLIG3 [257934] | R&D Systems | MAB2456 | 1:500 | N/A |
| Rabbit monoclonal anti-PAX6 [EPR15858] | Abcam | ab195045 | 1:300 | N/A |
| Rabbit polyclonal anti-PSD-95 (DLG4) | Abcam | ab18258 | N/A | 1:1000 |
| Mouse monoclonal anti-SATB2 [SATBA4B10] | Abcam | ab51502 | 1:25 | 1:100 |
| Rabbit monoclonal anti-SOX2 [D6D9] | Cell Signaling Technology | 3579 | 1:400 | N/A |
| Goat polyclonal anti-SOX2 | Santa Cruz | sc-17320 | N/A | 1:200 |
| Rabbit polyclonal anti-TBR1 | Abcam | ab31940 | 1:500 | 1:1000 |
| Mouse monoclonal anti-β-Tubulin III (TUJ1) [SDL.3D10] | Sigma-Aldrich | T8660 | 1:1000 | N/A |
| Donkey anti-Rabbit IgG (H + L) secondary antibody, Alexa Fluor 488 | Thermo Fisher Scientific | A-21206 | 1:500 | N/A |
| Donkey anti-Rat IgG (H + L) secondary antibody, Alexa Fluor 488 | Thermo Fisher Scientific | A-21208 | 1:500 | N/A |
| Donkey anti-Mouse IgG (H + L) secondary antibody, Alexa Fluor 594 | Thermo Fisher Scientific | A-21203 | 1:500 | N/A |
| Donkey anti-Rabbit IgG (H + L) secondary antibody, Alexa Fluor 594 | Thermo Fisher Scientific | A-21207 | 1:500 | N/A |
| IRDye 680RD goat anti-Rabbit | Li-Cor | 926-68071 | N/A | 1:15000 |
| IRDye 800CW goat anti-Mouse | Li-Cor | 926-32210 | N/A | 1:15000 |
| IRDye 800CW donkey anti-Goat | Li-Cor | 925-68074 | N/A | 1:15000 |

**Synaptosomal preparation.** Synaptic protein was extracted by manually dissociating cultured cells in 1x Syn-PER Reagent (Thermo Fisher Scientific) with added MS-SAFE Protease and Phosphatase Inhibitor (Sigma). Following low speed centrifugation to pellet cell debris (1200 g, 10 min, 4 °C) the supernatant was centrifuged at high speed to pellet synaptosomes (15,000 g, 20 min, 4 °C) which were resuspended in fresh Syn-PER Reagent. Protein concentration was determined using a DC Protein Assay (BioRad) quantified with the CLARIOstar microplate reader (BMG Labtech).

**Peptide affinity purification.** PDZ domain containing proteins were enriched from total protein extracts by peptide affinity purification. NMDA receptor subunit 2 C-terminal peptide "SIESDV" was synthesized (Pepceuticals) and fully dissolved in 90% v/v methanol + 1 M HEPES pH7 (both Sigma). Dissolved peptide was coupled to Affi-Gel 10 resin (Bio-Rad) that had been washed 3 times in methanol, followed by overnight room temperature incubation on a roller mixer. Unreacted NHS groups were subsequently blocked using 1 M Tris pH9 (Sigma) with 2 h room temp incubation on a roller mixer. The peptide bound resin was then washed 3 times with DOC buffer (1% w/v sodium deoxycholate; 50 mM Tris pH9; 1X MS-SAFE Protease and Phosphatase Inhibitor, all Sigma) and stored on ice until required. Total protein was extracted from dissociated cultured cells by incubating in DOC buffer for 1 h on ice with regular vortexing, cell debris was pelleted by high speed centrifugation (21,300 × g, 2 h, 4 °C) and the supernatant added to the previously prepared "SIESDV" peptide bound resin. After overnight 4 °C incubation on a roller mixer, the resin was washed 5 times with ice cold DOC buffer and the bound protein eluted by 15 min 70 °C incubation in 5% w/v sodium dodecyl sulfate (SDS, Sigma). The eluted protein was reduced with 10 mM TCEP and alkylated using 20 mM Iodoacetamide, trapped and washed on an S-trap micro spin column (ProtiFi, LLC) according to the manufacturer's instructions and protein digested using trypsin sequence grade (Pierce) at 47 °C for 1 h. Eluted peptides were dried in a vacuum concentrator and resuspended in 0.5% formic acid for MS analysis.

**Mass spectrometry analysis.** LC-MS/MS analysis was performed and data was processed and quantified according to[66]. Briefly, peptides were analyzed by nanoflow LC-MS/MS using an Orbitrap Elite (Thermo Fisher) hybrid mass spectrometer equipped with a nanospray source, coupled to an Ultimate RSLCnano LC System (Dionex) and Tune Plus for Orbitrap Elite (Thermo Fisher). Peptides were desalted on-line using a nano trap column, 75 μm I.D.X 20 mm (Thermo Fisher) and then separated using a 130-min gradient from 3 to 40% buffer B (0.5% formic

acid in 80% acetonitrile) on an EASY-Spray column, 50 cm × 50 µm ID, PepMap C18, 2 µm particles, 100 Å pore size (Thermo Fisher). The Orbitrap Elite was operated with a cycle of one MS (in the Orbitrap) acquired at a resolution of 60,000 at m/z 400, with the top 20 most abundant multiply charged (2+ and higher) ions in a given chromatographic window subjected to MS/MS fragmentation in the linear ion trap. An FTMS target value of 1e6 and an ion trap MSn target value of 1e4 were used with the lock mass (445.120025) enabled. Maximum FTMS scan accumulation time of 500 ms and maximum ion trap MSn scan accumulation time of 100 ms were used. Dynamic exclusion was enabled with a repeat duration of 45 s with an exclusion list of 500 and an exclusion duration of 30 s. Raw mass spectrometry data were analyzed with MaxQuant version 1.6.10.43[67]. Data were searched against a human UniProt (https://www.uniprot.org/) sequence database (downloaded December 2019) using the following search parameters: digestion set to Trypsin/P, methionine oxidation and N-terminal protein acetylation as variable modifications, cysteine carbamidomethylation as a fixed modification, match between runs enabled with a match time window of 0.7 min and a 20-min alignment time window, label-free quantification enabled with a minimum ratio count of 2, minimum number of neighbors of 3 and an average number of neighbors of 6. PSM and protein match thresholds were set to 0.1 ppm. A protein FDR of 0.01 and a peptide FDR of 0.01 were used for identification level cut-offs.

**CNV analysis**. Following manual dissociation of WT and *DLG2* KO hESC into DPBS, genomic DNA was extracted using the ISOLATE II Genomic DNA kit (Bioline). Following DNA amplification and fragmentation according to the associated Illumina HTS assay protocol samples were hybridized to an Infinium PsychArray v1.1 BeadChip (Illumina). The stained bead chip was imaged using the iScan System (Illuminia) and Genome Studio v2.0 software (Illumina) subsequently used to normalize the raw signal intensity data and perform genotype clustering. Final analysis for Copy Number Variation (CNV) was carried out with PennCNV software[68].

**RNA sequencing**. WT and *DLG2* KO cells were cultured to days 15, 20, 30 and 60 of cortical differentiation as described above (See 'Cortical differentiation'). Total transcriptome RNA was isolated from triplicate wells for all cell lines at each time point by lysing cells in TRIzol Reagent (Thermo Fisher Scientific) followed by purification with the PureLink RNA Mini Kit (Thermo Fisher Scientific). RNA quality control (QC) was performed with the RNA 6000 Nano kit analyzed using the 2100 Bioanalyzer Eukaryote Total RNA Nano assay (Agilent). cDNA libraries for sequencing were produced using the KAPA mRNA HyperPrep Kit for Illumina Platforms (Kapa Biosystems) and indexed with KAPA Single-Indexed Adapter Set A + B (Kapa Biosystems). Library quantification was by Qubit 1x dsDNA HS Assay kit (Thermo Fisher Scientific) and QC by High Sensitivity DNA kit analyzed using the 2100 Bioanalyzer High Sensitivity DNA assay (Agilent). Sequencing was performed using the HiSeq4000 Sequencing System (Illumina) with libraries split into 2 equimolar pools, each of which was run over 2 flow cell lanes with 75 base pair paired end reads and 8 base pair index reads.

All samples were modeled after the long-rna-seq-pipeline used by the PsychENCODE Consortium and available at https://www.synapse.org/#!Synapse:syn12026837. Briefly, the fastq files from Illumina HiSeq4000 were assessed for quality by using FastQC tool (v0.11.8)[69] and trimmed for adapter sequence and low base call quality (Phred score <30 at ends) with cutadapt (v2.3)[70]. The mapping of the trimmed reads was done using STAR (v2.7.0e)[71] and the BAM files were produced in both genomic and transcriptomic coordinates and sorted using samtools (v1.9)[72]. The aligned and sorted BAM files were further assessed for quality using Picard tools (v2.20.2)[73]. This revealed a high level of duplicate reads in day 30 KO2 samples (~72% compared to an average of 23% for other samples). These samples were removed prior to further analyses, which were thus performed on KO1 and WT samples for this timepoint. GRCh38.p12 was used as the reference genome and the comprehensive gene annotations on the primary assembly from Gencode (release 31) used as gene annotation. Gene and transcript-level quantifications were calculated using RSEM (v1.3.1)[74]. Both STAR and RSEM executions were performed using the psychENCODE parameters.

RSEM gene and isoform level estimated counts were imported using the tximport package (v1.12.3)[75]. Protein coding genes expressed (cpm ≥ 1) in at least 1/3 of the samples were taken forward for differential analyses of genes, transcripts and exons. Differential gene expression analysis was performed using the DESeq2 package (v1.24.0)[76] and differentially expressed genes were considered significant if their p value after Bonferroni correction was <0.05. Differential exon usage was analyzed using the DEXSeq pipeline[77]. Briefly, the GENCODE annotation.gtf file was translated into a .gff file with collapsed exon counting bins by using the dexseq_prepare_annotation.py script. Mapped reads overlapping each of the exon counting bins were then counted using the python_count.py script and the HTSEQ software (0.11.2)[78]. Finally, differential exon usage was evaluated using DEXSeq (v1.30)[77] and significant differences identified using an FDR threshold of 0.05. All the differential analyses were performed by using R (v3.6.1).

When analyzing differential gene expression in *DLG2*⁻/⁻ relative to WT, samples from KO1 and KO2 lines were combined i.e., for each timepoint a single differential gene expression analysis was performed, comparing expression in KO1 & KO2 samples against wild-type. To assess the impact of sample dropout at day 30, we investigated the similarity in gene expression between lines by clustering all KO1,

KO2 and WT samples (Supplementary Fig. 3a). At all 4 timepoints, all replicates from KO1 and KO2 cluster together: while KO2 samples from day 30 are not of sufficient quality to be used with confidence in further analyses, they are clearly similar to KO1 day 30 samples. We also performed differential expression analyses separately for each line (i.e., KO1 v WT and KO2 v WT) at all other timepoints. The overlap in expressed genes accounted for >98% of the genes expressed in each line and gene expression fold change was highly correlated between KO1 v WT and KO2 v WT (Spearman's $\rho^{day\ 15} = 0.67$, $\rho^{day\ 20} = 0.95$, $\rho^{day\ 60} = 0.75$). Overrepresentation odds ratios for GO terms also remain well correlated for significantly upregulated ($\rho^{day\ 15} = 0.70$, $\rho^{day\ 20} = 0.92$, $\rho^{day\ 60} = 0.67$) and down-regulated regulated ($\rho^{day\ 15} = 0.55$, $\rho^{day\ 20} = 0.95$, $\rho^{day\ 60} = 0.56$) genes. We noted that agreement between lines was greatest for day 20, which also lies close to the onset of neurogenesis and displays a high level of differential expression between KO and WT lines, comparable to that for day 30 (Fig. 1g). Further indicating a limited impact for sample dropout, phenotypes predicted by GO term analysis of differential expression at day 30 (deficits in neuron migration, morphology and action potential generation) were experimentally validated (Figs. 3 and 5); and all early neurogenic transcriptional programs identified in these data were shown to possess an identical profile of expression across human neurodevelopmental cell-types in vivo (Fig. 7).

**Transcriptional programs**. Genes were partitioned based upon their WT expression profiles as follows. Differentially expressed genes (Bonferroni P < 0.05) were first identified between pairs of timepoints (analyzing WT data only): genes upregulated in day 30 relative to day 20 (20–30$_{up}^{WT}$); genes upregulated in day 60 relative to day 30 (30–60$_{up}^{WT}$); and genes upregulated in day 60 relative to day 20 (20–60$_{up}^{WT}$). Early-transient, early-stable, early-increasing and late programs were then defined based upon the intersections of these gene-sets as shown in Fig. 4b.

**Human fetal cortex single-cell RNA sequencing data**. Single-cell RNA-Seq gene expression data from Nowakowski et al.[32] were downloaded from https://cells.ucsc.edu/?ds=cortex-dev. Cells corresponding to distinct neurodevelopmental cell-types (including cell-types at different stages of maturity) were identified and extracted, collating all cells from the corresponding in vivo cell clusters[32] as follows:

*Progenitors.*

RG (early): "RG-early"
RG: "RG-div1", "RG-div2", "oRG", "tRG", "vRG"
IPC: "IPC-div1", "IPC-div2"

*Transitioning.*

"IPC-nEN1", "IPC-nEN2", "IPC-nEN3"

*Cortical excitatory neurons.*

Newborn: "nEN-early1", "nEN-early2", "nEN-late"
Developing: "EN-PFC1", "EN-PFC2", "EN-PFC3", "EN-V1-1", "EN-V1-2", "EN-V1-3"

Cells with less than 5% of all protein-coding genes expressed (TPM ≥ 1) and genes expressed in less than 5% of cells were filtered out. The remaining dataset consisted of 2318 cells and 9239 protein-coding genes. Gene expression counts (TPM) were z-score normalized for each gene across all cells, then the average normalized expression score for each gene was calculated for each of the above cell-types. Over 80% of genes for each in vitro early neurogenic program (early-transient⁻/⁻, early-stable⁻/⁻, early-increasing⁻/⁻) and 55% (101 out of 182 genes) of the late-pyramidal$^{high}$ set were present in the in vivo data; all of these genes passed our filtering criteria. Taking the set of genes corresponding to each in vitro program, we calculated the mean and standard error of their gene-level averages in each in vivo cell-type (Fig. 7). For each program, the difference between successive neurodevelopmental cell-types/stages was calculated using a two-tailed Student's *t* test and *p* values Bonferroni-corrected for the 6 comparisons made: RG (early) v RG; RG v IPC; RG v Transitioning; IPC v Transitioning; Transitioning v Newborn neurons; and Newborn v Developing neurons.

To create a stringent set of fetal neuron expressed genes, we identified all genes expressed in at least 5% of Newborn cells or 5% of Developing cells (N$_{gene}$ = 9332). To create a more relaxed set, we identified all genes with non-zero expression in at least one Newborn or Developing cell (N$_{gene}$ = 16,431).

**Gene set construction**
*GO*. The Gene Ontology (GO) ontology tree was downloaded from OBO: http://purl.obolibrary.org/obo/go/go-basic.obo. Ontology trees were constructed separately for Molecular Function, Biological Process and Cellular Component using 'is_a' and 'part_of' relationships. GO annotations were downloaded from NCBI: ftp://ftp.ncbi.nlm.nih.gov/gene/DATA/gene2go.gz. Annotations containing the negative qualifier NOT were removed, as were all annotations with evidence codes IEA, NAS and RCA. Annotations were further restricted to protein-coding genes. Genes corresponding to each annotation term were then annotated with all

parents of that term, identified using the appropriate ontology tree. Finally, terms containing between 20 and 2000 genes were extracted for analysis.

*Regulator targets.* Predicted *TBR1* and *BCL11B* targets[42]: Transcription factor-target gene interactions identified by elastic net regression were downloaded from the PsychEncode resource website (http://resource.psychencode.org/#Derived) and predicted targets for *TBR1* and *BCL11B* extracted (interaction file: INT-11_ElasticNet_Filtered_Cutoff_0.1_GRN_1.csv). Gene symbols were mapped to NCBI/Entrez ids using data from the NCBI gene_info file.

TCF4 targets[40]: identifiers were updated using the gene_history file from NCBI.

FMRP targets[41]: NCBI/Entrez mouse gene identifiers were updated using the gene_history file from NCBI. Genes were then mapped from mouse to human using Homologene, restricting to protein-coding genes with a 1-1 mapping.

Direct *CHD8* mid-fetal promoter targets[37]: symbols (NCBI gene_info file) and locations (NCBI Build37.3) were used to map genes to NCBI/Entrez gene ids.

Indirect *CHD8* targets[36]: Ensembl ids were updated (Ensembl stable_id_event file) then mapped to current NCBI/Entrez ids using Ensembl and NCBI id cross-reference files. Taking genes with altered expression on *CHD8* shRNA knockdown, we removed those identified as direct *CHD8* targets in NPCs[36] or as *CHD8* mid-fetal promoter targets[37].

*Pyramidal CA1 'specific' (pyramidal^high^).* Mouse genes with specificity score >0 for pyramidal CA1 neurons were extracted from Supplementary Table 3 of Skene et al.[54] and mapped to human using the gene orthology file from NCBI (https://ftp.ncbi.nlm.nih.gov/gene/DATA/gene_orthologs.gz). To construct the pyramidal^high^ set we took the top 10% of 1-1 mapped genes ranked by specificity score ($N_{gene} = 1405$).

**Functional over-representation test (gene set overlap).** The degree of overlap between pairs of gene sets was evaluated using Fisher's Exact test, where the background set consisted of all genes expressed in either WT or *DLG2^−/−^* lines ($all^{WT+KO}$). This was used for GO terms; the analysis of regulator targets (Fig. 4e); and the overlap between LoFi genes and transcriptional programs (Fig. 6a). In order to identify a semi-independent subset of over-represented annotations from the output of GO term tests (Fig. 4f, Supplementary Data 5 and 7), we used an iterative refinement procedure. Briefly, we selected the gene set with the largest enrichment odds ratio; removed all genes in this set from all other over-represented annotations; re-tested these reduced gene-sets for over-representation in $30_{down}^{−/−}$ genes; then discarded gene-sets with $P \geq 0.05$ (after Bonferroni-correction for the number of sets tested in that iteration). This process was repeated (with gene-sets being cumulatively depleted of genes at each iteration) until there were no remaining sets with a corrected $P < 0.05$.

**Common variant association.** Common variant gene-set enrichment analyses were performed using the competitive gene-set enrichment test implemented in MAGMA version 1.07, conditioning on $all^{WT+KO}$ using the *condition-residualize* function. To test whether GO terms (Fig. 4f, Supplementary Data 8) or regulator targets (Fig. 4e) enriched in a specific program captured more or less of the SZ association in these programs than expected, a two-sided enrichment test was performed on term/target genes within the program, conditioning on $all^{WT+KO}$ and on all genes in the program. All other GWAS enrichment tests were one-sided. To test whether common variant enrichment differed between two gene-sets, we took the regression coefficient $\beta$ and its standard error $SE(\beta)$ for each gene-set from the MAGMA output file and compared $z = d/SE(d)$ to a standard normal distribution, where $d = \beta_1 − \beta_2$ and $SE(d) = \sqrt{[SE(\beta_1)^2 + SE(\beta_2)^2]}$. Gene-level association statistics for schizophrenia were taken from Pardiñas et al.[12]; those for ADHD[49], bipolar disorder[50] and Alzheimer's disease[53] were calculated using the MAGMA multi model, with a fixed 20,000 permutations for each gene. Prior to analysis, SNPs with MAF < 0.01 or INFO score <0.6 were removed from the bipolar GWAS, bringing it into line with the other datasets.

**Rare variant association.** The *de novo* LoF mutations for SZ analyzed here are described in Rees et al.[16]. *De novo* LoF mutations for NDD, ASD and unaffected siblings of individuals with ASD were taken from Satterstrom et al.[47]: these were re-annotated using Variant Effect Predictor[79] and mutations mapping to >2 genes (once readthrough annotations had been discarded) were removed from the analysis. A two-sided Poisson rate ratio test was used to evaluate whether the enrichment of *de novo* LoF mutations in specific gene-sets was significantly greater than that observed for all other expressed genes (using $all^{WT+KO}$). The expected rate of *de novo* LoF mutations in a set of genes was estimated using individual gene mutation rates[80].

**Migration assay.** Cells were cultured and differentiated to cortical projection neurons as previously described. Neuronal migration was measured during a 70-h period from day 40 by transferring cell culture plates to the IncuCyte Live Cell Analysis System (Sartorius). Cells were maintained at 37 °C and 5% $CO_2$ with 20X magnification phase contrast images taken of whole wells in every 2 h for the analysis period. The StackReg plugin[81] for ImageJ was used to fully align the resulting stacks of time lapse-images after which the cartesian coordinates of

individual neuronal soma were recorded over the course of the experiment, enabling the distance and speed of neuronal migration to be calculated. Data sets (Fig. 3f, g) were analyzed by unpaired two-tailed Student's *t* test.

**Morphology analysis.** Cells were differentiated to cortical projection neurons essentially as described and neuronal morphology assessed at days 30 and 70. To generate low density cultures for analysis, cells were passaged at either day 25 or 50 using 15-min Accutase solution (Sigma) dissociation followed by plating at 100,000 cells per well on 24 well culture plates. 72 h prior to morphology assessment cells were transfected with 500 ng pmaxGFP (Lonza) per well using Lipofectaime 3000 Reagent (Thermo Fisher Scientific) and Opti-MEM Reduced Serum Media (Thermo Fisher Scientific) for the preparation of DNA-lipid complexes. At days 30 or 70, cells were fixed in 4% paraformaldehyde (PFA, Sigma) in PBS for 20 min at 4 °C before mounting with Dako Fluorescence Mounting Medium (Agilent) and glass coverslips. Random fields were imaged using a DMI6000B Inverted microscope (Leica) and the morphology of GFP expressing cells with a clear neuronal phenotype quantified using the Neurolucida 360 v2.00.2 (MBF Bioscience) neuron tracing and analysis software package. Data sets (Fig. 3a–d) were analyzed by two-way ANOVA with post hoc comparisons using Bonferroni correction, comparing to WT controls.

**Electrophysiology.** Whole-cell patch clamp electrophysiology was performed on cells cultured on 13 mm round coverslips and the most morphologically mature neurons were patched in each culture; hence the most comparable subpopulation of cells from each genotype was compared. On day 20 of hESC differentiation, 250,000 human neural precursor cells from WT and KO hESCs were dissociated and plated on each PDL-coated coverslip in 30 µl diluted (20x) matrigel (Corning) together with 20,000 rat primary glial cells. Postnatal day 7–10 old Sprague-Dawley rats (Charles River) bred in-house were sacrificed via cervical dislocation and cortex was quickly dissected. Tissues were dissociated using 2 mg/ml papain and plated in DMEM supplemented with 10% Fetal bovine serum and 1% penicillin/streptomycin/Amphotericin B and 1x Glutamax (all Thermo Fisher Scientific). Microglia and oligodendrocyte precursor cells were removed by shaking at 500 rpm for 24 h at 37 °C. All animal procedures were performed in accordance with Cardiff University's animal care committee's regulations and the European Directive 2010/63/EU on the protection of animals used for scientific purposes. Plated cells were fed with BrainPhys medium (Stem cell Technologies) supplemented with 1x B27 (Thermo Fisher Scientific), 10 ng/ml BDNF (Cambridge Bioscience) and 200 µM ascorbic acid (Sigma). To stop the proliferation of cells, 1x CultureOne (Thermo Fisher Scientific) was supplemented from day 21. For postsynaptic current experiment, coverslips were transferred to a recording chamber (RC-26G, Warner Instruments) and perfused with HEPES Buffered Saline (HBS) (119 mM NaCl; 5 mM KCl; 25 mM HEPES; 33 mM glucose; 2 mM $CaCl_2$; 2 mM $MgCl_2$; 1 µM glycine; 100 µM picrotoxin; pH 7.4), at a flow rate of 2–3 ml per minute. Recordings were made using pipettes pulled from borosilicate glass capillaries (1.5 mm OD, 0.86 mm ID, Harvard Apparatus), and experiments were performed at room temperature (~20 °C). mEPSC recordings were made using recording electrodes filled with a Cs-based intracellular filling solution (130 mM $CsMeSO_4$; 8 mM NaCl; 4 mM Mg-ATP; 0.3 mM Na-GTP, 0.5 mM EGTA; 10 mM HEPES; 6 mM QX-314; with pH 7.3 and osmolarity ~295 mOsm). Cells were voltage clamped at −60 mV using a Multiclamp 700B amplifier (Axon Instruments). Continuous current acquisition, series resistance and input resistance were monitored and analyzed online and offline using the WinLTP software[82] (http://www.winltp.com). Only cells with series resistance <25 MΩ with a change in series resistance <10% from the start were included in this study. Data were analyzed by importing Axon Binary Files into Clampfit (version 10.6; Molecular Devices). A threshold function of >12 pA was used to identify mEPSC events, which were then subject to manual confirmation. Results were outputted to SigmaPlot (version 12.5, Systat Software), where analysis of peak amplitude and frequency of events was performed.

The current clamp was used to record resting membrane potential (RMP) and action potentials (AP). Data were sampled at 20 kHz with a 3 kHz Bessel filter with MultiClamp 700B amplifier. Coverslips were transferred into the recording chamber maintained at RT (20-21 °C) on the stage of an Olympus BX61W (Olympus) differential interference contrast (DIC) microscope and perfused at 2.5 ml/min with the external solution composed of 135 mM NaCl, 3 mM KCl, 1.2 mM $MgCl_2$, 1.25 mM $CaCl_2$, 15 mM D-glucose, 5 mM HEPES (all from Sigma), and pH was titrated to 7.4 by NaOH. The internal solution used to fill the patch pipettes was composed of 117 mM KCl, 10 mM NaCl, 11 mM HEPES, 2 mM $Na_2$-ATP, 2 mM Na-GTP, 1.2 mM $Na_2$-phosphocreatine, 2 mM $MgCl_2$, 1 mM $CaCl_2$ and 11 mM EGTA (all from Sigma), and pH was titrated to 7.3 by NaOH. The resistance of a patch pipette was 3–9 MΩ and the series resistance component was fully compensated using the bridge balance function of the instrument. The RMP of cells was recorded immediately after breaking into the cells in gap free mode. A systematic current injection protocol (duration, 1 s; increment, 20 pA; from −60 pA to 120 pA) was applied to the neurons held at −60 mV to evoke APs. Input resistance ($R_{in}$) was calculated by $R_{in} = (V_i − V_m)/I$, where $V_i$ is the potential recorded from −10 pA current step. The AP properties are measured by the first over shooting AP. Further analysis for action potential characterization was carried out by Clampfit 10.7 software (Molecular Devices).

**Statistics and reproducibility**. Unless specifically stated in each methodology section, GraphPad Prism (version 8.3.0) was used to test the statistical significance of the data and to produce the graphs. Stars above bars in each graph represents Bonferroni-corrected post hoc tests, $*P < 0.05$; $**P < 0.01$; $***P < 0.001$; $****P < 0.0001$ vs. WT control. All phenotypic validation results were from a minimum of two independent differentiations. Within a given differentiation triplicate samples were used per cell line at each time point investigated.

**Reporting summary**. Further information on research design is available in the Nature Research Reporting Summary linked to this article.

## Data availability

RNAseq data generated by this study have been deposited in the Gene Expression Omnibus (GEO) archive with accession number GSE172199. The mass spectrometry data generated in this study have been deposited in the ProteomeXchange Consortium via the PRIDE partner repository with the dataset identifier PXD029526. Human proteome reference sequences are available in UniProt (https://www.uniprot.org/). Processed gene annotation data (GO terms, LoFi genes, BCL11B, TBR1, FMRP, TCF4, CHD8 target genes and Pyramidal^high set) are available in the accompanying Supplementary Software zip file. The human fetal single cell RNAseq data used in this study are available in the UCSC Cell Browser repository (https://cells.ucsc.edu/?ds=cortex-dev). GWAS data used in this study are available in the Psychiatric Genomics Consortium download database (https://www.med.unc.edu/pgc/download-results/) under accession codes: 29483656 (SZ, https://figshare.com/articles/dataset/scz2018clozuk/14681220), 31043756 (BP, https://figshare.com/articles/dataset/bip2019/14671998), 30478444 (ADHD, https://figshare.com/articles/dataset/adhd2019/14671965), 30804558 (ASD, https://figshare.com/articles/dataset/asd2019/14671989). The MDD GWAS data are available under restricted access due to inclusion of data from 23andMe, for which permission must be obtained separately. Access can be obtained by qualified researchers under an agreement with 23andMe that protects the privacy of the 23andMe participants; see cited study for details (https://www.nature.com/articles/s41588-018-0090-3). Summary statistics for the MDD GWAS excluding the 23andMe data are available from the Psychiatric Genomics Consortium download database (https://www.med.unc.edu/pgc/download-results/) under accession code 29700475 (https://figshare.com/articles/dataset/mdd2018/14672085). The IQ GWAS data used in this study are available from (https://ctg.cncr.nl/software/summary_statistics/) as a compressed file (SavageJansen_IntMeta_sumstats.zip). The AD GWAS data used in this study are available from the authors of the original study (https://www.nature.com/articles/ng.2802). The SZ de novo rare variant data used in this study are available from the cited study (https://www.nature.com/articles/s41593-019-0565-2). The ASD and NDD de novo rare variant data used in this study are available in Supplementary Tables 1 and 4 of the cited study (https://www.cell.com/cms/10.1016/j.cell.2019.12.036/attachment/44aca411-6be3-4158-b1d6-6b339d60136d/mmc1.xlsx, https://www.cell.com/cms/10.1016/j.cell.2019.12.036/attachment/45f1baed-a2ec-4128-be82-a50ae08adc38/mmc4.xlsx). The GRCh38.p12 genome reference sequence and gene annotation data used for the bulk RNAseq analysis are available from GENCODE (https://ftp.ebi.ac.uk/pub/databases/gencode/Gencode_human/release_31/gencode.v31.primary_assembly.annotation.gtf.gz, https://ftp.ebi.ac.uk/pub/databases/gencode/Gencode_human/release_31/GRCh38.primary_assembly.genome.fa.gz). Source data are provided with this paper.

## Material availability

H7 hESC line was purchased from WiCell. DLG2 knockouts and WT sister line created from H7 are available from the corresponding author upon request, provided the purpose of the transfer is within the MOU and SLA from WiCell.

## Code availability

All publicly available software utilized is noted in Methods. Analysis scripts are provided with this paper in Supplementary Software and as a citeable GitHub repository (https://zenodo.org/record/5729267#.YaeoXtDP2Uk)[83].

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

## Acknowledgements

This work was supported by Wellcome Trust Strategic Award (100202/Z/12/Z, A.H. & M.O.), MRC programme grant (G08005009, M.O.D., M.O. & A.P.), MRC Centre grant (MR/L010305/1, M.O.D., M.O. & A.P.), Waterloo Foundation 'Changing Minds' programme (506926, E.S.) and start-up funding (AH1132S012, AH1132S204, E.S.) from the Neuroscience and Mental Health Research Institute, Cardiff University. We acknowledge excellent technical support for RNA sequencing from Joanne Morgan (MRC Centre) and MS analysis from Lydia Kiesel (University of Sheffield) and assistance in morphology tracing from Sophie Pocklington. We appreciate excellent general lab support from Emma Dalton, Trudy Workman and Olena Petter. We thank Prof. Meng Li for her advice and Dr. Claudia Tamburini for technical support in the initial stages of the project and Profs. Yves Barde and Lesley Jones for helpful comments on the manuscript and Emily Adair for providing rat primary glial cells. We thank the International Genomics of Alzheimer's Project (IGAP) for providing summary results data for AD common variant analysis. The investigators within IGAP contributed to the design and implementation of IGAP and/or provided data but did not participate in analysis or writing of this report. IGAP was made possible by the generous participation of the control subjects, the patients, and their families. The i–Select chips was funded by the French National Foundation on Alzheimer's disease and related disorders. EADI was supported by the LABEX (laboratory of excellence program investment for the future) DISTALZ grant, Inserm, Institut Pasteur de Lille, Université de Lille 2 and the Lille University Hospital. GERAD was supported by the Medical Research Council (Grant no. 503480), Alzheimer's Research UK (Grant no. 503176), the Wellcome Trust (Grant no. 082604/2/07/Z) and German Federal Ministry of Education and Research (BMBF): Competence Network Dementia (CND) grant no. 01GI0102, 01GI0711, 01GI0420. CHARGE was partly

supported by the NIH/NIA grant R01 AG033193 and the NIA AG081220 and AGES contract N01–AG–12100, the NHLBI grant R01 HL105756, the Icelandic Heart Association, and the Erasmus Medical Center and Erasmus University. ADGC was supported by the NIH/NIA grants: U01 AG032984, U24 AG021886, U01 AG016976, and the Alzheimer's Association grant ADGC–10–19672. We thank the research participants and employees of 23andMe for the sharing of summary statistics for MDD common variant analysis.

## Author contributions

Conceptualization, A.P., E.S. Design of Methodology, M.C., A.E., D.B., A.P., E.S. Software/Data curation, B.S., D.D.A., M.C., A.P., E.S. Formal analysis/Investigation, B.S., D.D.A., M.C., T.S., E.R., Y.Z., G.C., S.L., A.F.P., D.W., A.P., E.S. Writing – Original Draft, B.S., D.D.A., M.C., Y.Z., D.W., A.P., E.S. Writing – Review & Editing, B.S., D.D.A., M.C., E.R., S.L., A.F.P., W.G., M.O.D., M.O., A.E., D.B., D.W., A.P., E.S. Visualization, B.S., D.D.A., M.C., Y.Z., D.W., A.P., E.S. Supervision, A.P., E.S. Funding acquisition, A.H., M.O.D., M.O., A.P., E.S.

## Competing interests

D.D.A., Y.Z., A.H., W.G., M.O.D., M.O., A.P. are supported by a collaborative research grant from Takeda (Takeda played no part in the conception, design, implementation, or interpretation of this study). The other authors report no financial relationships with commercial interests. The remaining authors declare no competing interests.

## Additional information

**Peer review information** *Nature Communications* thanks Zhexing Wen and the other anonymous reviewer(s) for their contribution to the peer review this work. Peer reviewer reports are available.

