## [Peer Review File · Nature Communications]

Reviewers' Comments:

Reviewer #1:

Remarks to the Author:

The manuscript by Sanders et al., (NCOMMS-20-20995-T) in essence, examines an excitatory synapse scaffold gene, *DLG2*, and the impact of knocking out *DLG2* in hESCs to investigate how this loss-of-function intolerant (LoFi) gene highly expressed in the mid-foetal brain disrupts developmental pathways (neuronal migration, morphology, generation of action potentials) in vitro.

There is clear, growing evidence that rare, penetrant mutations in synapse genes including *DLG2*, are involved in NDDs. In this paper, the authors show the novel generation and characterisation of the *DLG2* knockout in hESCs. I believe previous work to date has only examined *DLG2* knockout phenotypes in mice. The data presented are interesting; however, a major concern is the overstretch of how this work is framed, and the leaps made with regards to the interpretation. Based on the experimental data provided, this reviewer was confused with the what this paper robustly shows compared to that claimed. The direct evidence supporting the idea that "DLG2 links cell-surface receptors to signal transduction molecules driving the activation of neurogenic transcriptional programs", thus the "data reveal a major aetiological role for common neurodevelopmental pathways in neuropsychiatric disorders and cognition" was not clear or convincing.

hEPSCs remain a valuable tool to investigate patient derived variants but are compromised by the immaturity of the culture system. While the focus on hEPSCs is understandable given the focus on *DLG2*, it's difficult to fully appreciate in isolation how these data robustly recapitulate the developing brain that receives influence from other cell types (interneurons, astrocytes, microglia) that have been shown to play pivotal roles in shaping circuit function and dysfunction in NDDs. Extending components of the current analysis to assess the same developmental measures in a murine model in vivo would have strengthened the interpretations. Additionally, could a 3D organoid culture system be employed in parallel to enhance validation towards more closely mimicking human brain development and physiology?

The results and figures are well presented. The manuscript (in particular the Abstract, Introduction and Discussion) would however benefit from the text being tightened and refocused for flow and clarity, to help integrate rationales in the context of what the results convey.

Reviewer #2:

Remarks to the Author:

DLG2 Knockout reveals neurogenic transcriptional programs

This is an exciting paper which tackles the issue of when the genetic insult underlying schizophrenia occurs, and how this relates to the effect of using rare disease associated variants. They use GWAS heritability analyses in an informative manner to dissect the developmental time points of greatest importance. They show that there is a similarity between the changes found in cultures and the developing human brain. The paper largely focuses on the *DLG2* associated changes (which I actually found to be the weaker part of the story) rather than the general developmental changes.

Major Issues:

- To my mind the enrichment of SZ heritability in genes upregulated between P20->P30 is actually the key result of the paper, and more important than the *Dlg2* findings. The current presentation of the results suggests that the schizophrenia enrichment is inherently a developmental change. An alternative way of looking at it is that a change in cell type/state has occurred: neural progenitors have transitioned into neurons, hence everything upregulated at/after day 30 is neuron specific (and so expected to be enriched for SZ heritability). It would strengthen the results to show that the upregulated genes are still enriched when conditioned on genes specific to adult

neurons, as this would suggest the enrichment is really related to development. They could also use some kind of cell type deconvolution approach to show whether the DE represents a change in cell type.

- DLG2 is not significantly differentially expressed at most time points between wildtype and knockouts (Supp Table 3) which is somewhat concerning: how reliable is the DE if it doesn't detect a gene knockout? They should report the base mean and DE for DLG2 in a main figure.
- There is a very significant difference in the number of CTIP2 positive cells between KO1 and KO2 at P40, P50 and P60. This indicates that the age at which the cultures reach maturity is a highly variable process. As CTIP2 density (or the neuronal maturity that it represents) appears to underlie the SZ associated differential expression, this presents an issue with claiming the changes are really DLG2 associated: perhaps it's just a fluke relating to how the cultures were handled. From reading the methods, I cannot quite tell if the wild type cultures were handled in exactly the same way as the genetically modified cultures: were the wild type cultures also nucleofected and FACS sorted prior to plating? My impression is that no attempt was made to pseudo-nucleofect the control cultures in the same manner. If this process stresses the cells then it seems plausible that this could have stressed the cells and delayed their differentiation as a result. If this is the case, would it be possible to try testing whether the process affects the CTIP2 density as shown in SuppFig6? Recognising that getting any experiments done at present, at the very least Supplemental Figure 6 should be included as a main figure, the significance of the difference between the two cultures stated and this possibility clearly noted.
- All bar plots used to show differences between groups, e.g. Fig 2c,d,e,f,g,h and Fig3a,b,c,d,f,g should be replaced with boxplots, jitter plots or ideally, estimation plots (<https://www.nature.com/articles/s41592-019-0470-3>).
- No data / code availability sections. The descriptions of some sections of the computational analysis are too vague; sharing the code would clear up any ambiguities. All scripts relating to differential expression etc should just be posted on GitHub or similar. The datasets were not made available to reviewers. How the differential expression was obtained for the single-cell RNA-seq is particularly unclear: did they combine data from all cell types at a particular time point?

Minor Issues:

- They state that "this suggests that a significant proportion of SZ common variants may contribute to disease via the disruption of neurodevelopmental pathways harbouring a concentration of LoFi genes". They state this because SZ genes are enriched for LoFi genes, but LoFi genes are broadly important for nervous system function in adulthood not just neurodevelopment, so this is a misleading statement.
- Would be useful to include a UMAP / tSNE / PCA / hclust plot of the datasets to get a sense of where the variation is
- They are not clear about exactly how differential expression was calculated. They state that expression data from the two DLG2^{-/-} lines were pooled: does this mean the model formula was just $\text{expression} \sim \text{genotype}$? Could they also share differential expression results between the two DLG2^{-/-} lines (and if anything is DE whether that is enriched for schiz heritability at any time points).
- In SuppFig6d,e,f at P50 for KO1+KO2, CTIP2 expression is 50% of WT... but for KO1 it's ~10% and for KO2 it's ~2%. So how does 10% + 2% get to 50%?
- They say "captured the signal". Should be "captured the heritability enrichment" or similar.
- They say "while complexes regulating postsynaptic information processing have been robustly implicated in SZ, this represents the first evidence that electrical properties underlying information transmission are also disrupted". It's a bit much to say this on the basis of a GO enrichment.

- They say "In summary, DLG2^{-/-} neurons have a reduced ability to fire APs and produce less mature APs." This should be tempered with "at day 50" and should acknowledge that it's likely to just be delayed rather than a permanent deficiency.
- A supplementary table should be included with the differential expression results from the single cell analysis
- They note in the methods that the day 30 KO2 samples were dropped: could they be a bit more explicit that this meant that only KO1 samples were used at day 30.
- On page 15 they report for day 30 the correlation between WT vs KO, performed for each cell line separately; it's remarkably high. Could they report the same values for other days?

Reviewer #3:

Remarks to the Author:

The manuscript by Sanders et al. investigates the neurodevelopmental role of a synaptic protein, DLG2, during directed differentiation of human pluripotent stem cells towards the dorsal telencephalon. DLG2 has been implicated in the etiology of Schizophrenia and the authors demonstrate that loss of DLG2 function alters a neurogenic transcriptional program that is similarly found in other neuropsychiatric/neurodevelopmental diseases suggesting this transcriptional program is a point of convergence. Specifically, the authors uncovered a cell type specific delay in excitatory deep layer neuron generation (marked by BCL11B/CTIP2 expression) during cortical neurogenesis. DLG2 knockout neurons were shown to have decreased migration, neuronal complexity and functionality. Overall this is a well-presented manuscript and provides evidence linking DLG2 as a major factor of convergence in neurodevelopmental disorders. I do have some concerns on the regionalization and cell type specificity from the differentiations performed and how it might impact the interpretation and the overall conclusions of the data summarized below:

1. The authors identify the day 30 timepoint (start of neurogenesis) on their differentiation where a particular enrichment for schizophrenia related genes deviated from wildtype controls (Figure 2B and 4B) and surprisingly is not enriched in the late stage (day 60). This is a very interesting point since the culture should be enriched for immature neurons and neural progenitors and DLG2 may be involved in other processes in addition to the synapse. However, looking at the gene expression changes in Supplemental Table 3, it appears that several genes such as DLX1, DLX2, GBX2, NKX2.1 and FOXP1 are significantly and differentially expressed at this particular timepoint between WT and mutant.

It would be important for the authors to investigate and confirm the cell type specificity of the cells generated other than simply using TBR1, CTIP2 and SATB2 as markers for cortical excitatory neurons. The expression of the DLX, GBX and NKX genes suggests that it's highly probable that the cells at this particular timepoint were ventralized. TBR1 can be expressed in the ventral thalamus, CTIP2 in the striatum and SATB2 has been described as expressed in the hypothalamus. Excluding these brain regions would greatly strengthen the conclusions of the author's comparative analysis.

2. Somewhat related to the point above, neural progenitors in culture typically generate the cortical layers in a sequential and time dependent fashion. The data the authors present in Figure 1F show a modest increase in TBR1⁺ cells over time suggesting that the majority of these neurons are born on day 30. Subsequently, there appears to be no change in the generation of SATB2⁺ cells in the same time course (Supplemental Figure 5H) suggesting that those neurons are either not born yet (based on the low percentage of cells), or already established at day 30. The authors need to show supporting evidence that the expression of SATB2 in their cultures indeed represent the cell type they intend. Or they will need to modify their conclusion: "A similar analysis of layer markers TBR1, CTIP2 and SATB2 revealed a significant decrease in CTIP2⁺ cells but a comparable proportion of TBR1⁺ and SATB2⁺ neurons for all timepoints investigated (Fig. 2d, e, g-i, Supplementary Fig. 5)."

3. The data that indicates a delay in the generation of CTIP2⁺ neurons due to the loss of DLG2

needs additional experiments to support this conclusion. The authors show that the generation of TBR1 and SATB2 is roughly equivalent between WT and KO, but CTIP2+ neurons are dramatically lost/delayed in the cells derived from the KO line. Eventually, CTIP2+ cells "recover" with time in the KO line. Are the CTIP2+ cells actually coming from cortical patterned NSCs? Lineage tracing or clonal labeling of the starting radial glial cells will confirm that the DLG2 KO radial glia will produce the cortex in an "out of order" manner. Alternatively, sorting out the cortical radial glial cells from these timepoints and triggering the differentiation would help to address this question as well.

4. In both the morphology and the electrophysiology studies, it is unclear how the authors decided which neurons to assay. Given the heterogeneity of the neurons in the dish at day 50 (Ephys) and d70 (morphology), and the potential delay in neurogenesis of CTIP2+ neurons from the DLG2 KO, could some of the neurons be at different stages in development? How did the authors decide on the criteria used to distinguish the mature and immature neurons rather than an arbitrary timepoint? More details of this would be beneficial towards the interpretation of their data.

We thank the reviewers for their comments, which have improved and clarified the manuscript. In addition to new experimental data and analyses discussed further below, we have taken the opportunity to filter out low frequency (MAF < 0.01) and poorly imputed (INFO < 0.6) variants from the Bipolar disorder GWAS, bringing it into line with the other GWAS datasets. Below we address each of the points raised in turn.

Reviewer #1

The manuscript by Sanders et al., (NCOMMS-20-20995-T) in essence, examines an excitatory synapse scaffold gene, DLG2, and the impact of knocking out DLG2 in hESCs to investigate how this loss-of-function intolerant (LoFi) gene highly expressed in the mid-foetal brain disrupts developmental pathways (neuronal migration, morphology, generation of action potentials) in vitro.

1. There is clear, growing evidence that rare, penetrant mutations in synapse genes including DLG2, are involved in NDDs. In this paper, the authors show the novel generation and characterisation of the DLG2 knockout in hESCs. I believe previous work to date has only examined DLG2 knockout phenotypes in mice. The data presented are interesting; however, a major concern is the overstretch of how this work is framed, and the leaps made with regards to the interpretation. Based on the experimental data provided, this reviewer was confused with the what this paper robustly shows compared to that claimed. The direct evidence supporting the idea that “DLG2 links cell-surface receptors to signal transduction molecules driving the activation of neurogenic transcriptional programs”, thus the “data reveal a major aetiological role for common neurodevelopmental pathways in neuropsychiatric disorders and cognition’ was not clear or convincing.

Response: We apologise if the discussion and interpretation of our results were unclear. We have substantially re-written the discussion, making the distinction between what we have shown and what we hypothesise as transparent as possible.

To briefly review, our main findings concern the identification of 3 sets of genes activated during early neurogenesis. Each set has a distinct functional composition and characteristic mRNA expression profile over neurogenesis (observed in both *in vitro* and *in vivo* data, see revised Figure 7). These distinct transcriptional programs are enriched for genetic variants influencing cognitive function and conferring risk for numerous neuropsychiatric disorders, displaying robust association in 9 independent genetic datasets. The strength of the genetic findings allows us to conclude that these programs play an important aetiological role across a wide spectrum of psychiatric disorders. In addition, the fact that expression of these programs is significantly perturbed in *DLG2* KO lines allows us to investigate the phenotypic consequences of their dysregulation – phenotypes that we predict based on the known functional roles of genes comprising these programs. The experimental data reveal clear differences in the expression of cell-type identity and the morphology, migration and electrophysiological properties of new-born neurons, validating our predictions and showing that *DLG2* plays an important role in regulating these processes. Within programs, schizophrenia common variant association was shown to be distributed across functional sets of genes regulating developmental processes perturbed in *DLG2* KO lines, implicating these processes/phenotypes in disease pathogenesis.

In the revised manuscript, we extend the genetic analysis of functional sets within each program to the other disorders studied (Supplementary Tables 7 & 8). This indicates that, within each

program, the perturbation of multiple molecular processes is likely to contribute to any one disorder, although the relative importance of different processes can vary between disorders.

We do not claim to show that '*DLG2 links cell-surface receptors to signal transduction molecules driving the activation of neurogenic transcriptional programs*', we simply propose this as a hypothetical model based on the known role of DLG proteins in the assembly and function of signal transduction complexes. As noted above we have sought to clarify this in the revised manuscript. In reference to the point raised here, the statement '*... we hypothesised that DLG2-scaffolded complexes may also regulate early neurodevelopmental transcriptional programs...*' has been removed from the introduction. We agree that this is potentially confusing, setting up a reasonable expectation that our data will address this hypothesis in full. This requires a separate study to identify and experimentally manipulate DLG2 interactors, which is not our aim here. The statement that the '*data reveal a major aetiological role for common neurodevelopmental pathways in neuropsychiatric disorders and cognition*' is based on our *in vitro/in vivo* gene expression and human genetic analyses, as indicated above, irrespective of the mechanism of DLG2 action.

2. hEPSCs remain a valuable tool to investigate patient derived variants but are compromised by the immaturity of the culture system. While the focus on hEPSCs is understandable given the focus on DLG2, it's difficult to fully appreciate in isolation how these data robustly recapitulate the developing brain that receives influence from other cell types (interneurons, astrocytes, microglia) that have been shown to play pivotal roles in shaping circuit function and dysfunction in NDDs. Extending components of the current analysis to assess the same developmental measures in a murine model in vivo would have strengthened the interpretations. Additionally, could a 3D organoid culture system be employed in parallel to enhance validation towards more closely mimicking human brain development and physiology?

Response: It is true that our *in vitro* cultures do not capture interactions that occur between cortical excitatory neurons and interneurons/astroglia as neuronal circuits form and mature. This would be a valid criticism if we were studying circuit formation and function, but we are not. Our study concerns cellular pathways activated during the very earliest stages of neurogenesis as neural precursors start to produce new-born, immature excitatory neurons; these stages are well modelled by 2D culture systems (Chambers et al., Nat Biotech, 2009; van de Leemput et al., 2014, Neuron). When investigating the effects of pathway dysregulation in *DLG2* KO lines, we focus on the intrinsic properties of individual new-born excitatory neurons: their morphology, electrophysiology and ability to migrate. Interaction with interneurons only occurs once both pyramidal neurons and interneurons have migrated into the developing cortex, while astroglia only begin to be generated once cortical neurogenesis is complete (Molnár et al., J Anat 2019). While it would of course be interesting to investigate the impact of *DLG2* KO at these much later timepoints, this lies outside the scope of the current study. What our data clearly show is that risk variants impact brain development during very early neurogenesis, long before the onset of circuit formation.

As noted above, our main findings concern 3 transcriptional programs activated during early neurogenesis, each with a unique gene expression time-course. We agree that our data would be strengthened by showing these programs to be active during cortical excitatory neurogenesis *in vivo* – preferably in human tissue rather than murine models. In the revised manuscript, we use single-cell RNAseq data from human foetal brain samples (Nowakowski et al., Science 2017) to evaluate the expression of these programs in neurodevelopmental cell-types spanning cortical

excitatory neurogenesis. For all 3 programs, human *in vivo* gene expression in individual neurodevelopmental cell-types recapitulates the developmental profile seen in our bulk RNAseq data *in vitro* (see revised Figure 7).

3. *The results and figures are well presented. The manuscript (in particular the Abstract, Introduction and Discussion) would however benefit from the text being tightened and refocused for flow and clarity, to help integrate rationales in the context of what the results convey.*

Response: Abstract, introduction and discussion have been revised, with a clearer focus on our main findings.

Reviewer #2

DLG2 Knockout reveals neurogenic transcriptional programs

This is an exciting paper which tackles the issue of when the genetic insult underlying schizophrenia occurs, and how this relates to the effect of using rare disease associated variants. They use GWAS heritability analyses in an informative manner to dissect the developmental time points of greatest importance. They show that there is a similarity between the changes found in cultures and the developing human brain. The paper largely focuses on the DLG2 associated changes (which I actually found to be the weaker part of the story) rather than the general developmental changes.

Major Issues:

1. *To my mind the enrichment of SZ heritability in genes upregulated between P20->P30 is actually the key result of the paper, and more important than the Dlg2 findings. The current presentation of the results suggests that the schizophrenia enrichment is inherently a developmental change. An alternative way of looking at it is that a change in cell type/state has occurred: neural progenitors have transitioned into neurons, hence everything upregulated at/after day 30 is neuron specific (and so expected to be enriched for SZ heritability). It would strengthen the results to show that the upregulated genes are still enriched when conditioned on genes specific to adult neurons, as this would suggest the enrichment is really related to development. They could also use some kind of cell type deconvolution approach to show whether the DE represents a change in cell type.*

Response: As should be clear from our responses to Review #1 above, we agree that the identification of neurodevelopmentally regulated gene sets robustly enriched for genetic association (not just for schizophrenia) is the most impactful and substantial finding. Uncovering a developmental role for *DLG2* is also important and cannot be disentangled from the other findings. The study of *DLG2* KO lines not only led to the identification of developmentally regulated gene sets, but also allowed us to test and validate predicted phenotypes based on computational analysis of these sets: we believe this is a major strength of the study.

To our minds, at the cellular level a developmental change is by definition a change in cell type or state and we investigate the changes in *DLG2* KO lines in precisely these terms. We show that there is no difference in the rate at which progenitors transition into neurons (i.e. no change in relative cell-type abundance), but that there is delayed maturation and expression of cell-type identity within new-born neurons (i.e. altered cell state). Furthermore, our new analysis of *in vivo* cell-type specific gene expression (see revised Figure 7) clearly indicates that early-transient^{-/-}

genes are normally up-regulated in radial glia as they mature, representing a change in cell state. The other programs are expressed at high levels in neurons, with higher early-increasing^{-/-} expression being a marker of more mature neurons – another change in cell state. At the risk of over-labouring the point: neurons are still being born at the same rate in *DLG2* KO lines, it is the activation of gene expression programs that is being delayed, altering cells' internal state. Given the new analysis of *in vivo* single-cell RNAseq data outlined above (revised Figure 7) and the estimation of neuron and progenitor abundance during the course of differentiation by immunocytochemistry (Figure 2 and Supplementary Figure 7), there is no reason to pursue *in silico* deconvolution of our bulk RNAseq data.

We have been extremely careful to ensure that our results are not driven by general neuronal enrichment for schizophrenia genetic risk. As we state at the very beginning of our genetic analyses (p4):

“...the set of all genes expressed at one or more timepoint in *DLG2*^{-/-} or WT lines (all^{WT+KO}) was highly enriched for common variant association ($P = 2.1 \times 10^{-17}$) reflecting the neural lineage of these cells. We therefore tested genes up- and down-regulated at each timepoint for genetic association conditioning on all^{WT+KO} using the strict condition-residualize procedure (all subsequent GWAS enrichment tests were conditioned on all^{WT+KO} in the same way).”

All of our genetic analyses control for the background level of association present in expressed neural/neuronal genes, thus all of the genetic associations we report are significant enrichments over and above any generic ‘neuronal’ signal. We explicitly show that association enrichment is not a generic property of genes normally up-regulated at/after day 30 (*Distinct transcriptional programs regulated by DLG2 are enriched for common SZ risk alleles*, p5): genes up-regulated on day 30 that are not down-regulated on *DLG2* KO are not enriched for association; and genes only up-regulated after day 30 (the ‘late’ set) are also not enriched for association (p5 and Figure 4 b-c).

Our data provide compelling evidence that risk genes belonging to early neurogenic programs contribute to early developmental deficits with psychopathological consequences, irrespective of whether they are expressed in adult neurons. Numerous studies have reported brain structural and cytoarchitectural changes associated with schizophrenia (e.g. see review by Harrison, Brain 1999): the programs we identify are enriched for genes contributing to neuron migration and morphology and their dysregulation in *DLG2* KO lines leads to corresponding phenotypes. We also see strong enrichment for genes regulating axon guidance and synapse structure/function indicating that genetic variation is also likely to impact network formation and synaptogenesis, although as noted above validation of this requires a dedicated study of its own. Large-scale genotyping studies have revealed substantial genetic overlap between schizophrenia and congenital/childhood-onset disorders including intellectual disability/severe neurodevelopmental delay (ID/NDD), autism spectrum disorders (ASD) and attention-deficit/hyperactivity disorder (ADHD) (Purcell et al., 2009; Sebat et al., Trends Genet 2009; Williams et al., Lancet 2010; Girirajan et al., PLoS Genet 2012; Anttila et al., Science 2018; Grove et al., Nat Genet 2019; Demontis et al., Nat Genet 2019). We show that early neurogenic programs are highly enriched for variants contributing to all of these pre-natal/early post-natal onset conditions. Given the weight of this evidence, a far stronger argument can be made that – for genes with both early developmental and adult expression – the burden of proof lies in showing that their function in mature neurons also contributes to disease, in addition to their developmental role.

2. *DLG2* is not significantly differentially expressed at most time points between wildtype and knockouts (Supp Table 3) which is somewhat concerning: how reliable is the DE if it doesn't detect a gene knockout? They should report the base mean and DE for *DLG2* in a main figure.

Response: Gene-level transcript abundance is a poor marker of gene knockout. Premature stop codons and frameshift mutations (such as those introduced into the *DLG2* gene in our KO lines) do not prevent mRNA being produced: full length *DLG2* mRNAs will still be produced but will then be degraded without being translated into protein. At the gene level, differential mRNA expression can potentially detect significant changes in the rate of mRNA degradation in KO lines, but not whether *DLG2* protein is or is not produced. Only the absence of protein expression is accepted as conclusive evidence for knockout of a protein-coding gene, with mRNA data playing at most a supporting role. We therefore cite the absence of *DLG2* protein in KO lines as evidence for the effective KO of *DLG2* ('*Knockout generation and validation*', p3), alongside supporting transcript and exon quantification that shows a significant reduction in the major *DLG2* transcripts and PDZ-encoding exons in KO lines at days 30 and 60 (Supplementary Figure 3, revised Supplementary Figure 4).

Reporting mRNA expression for *DLG2* would not be informative, for the reasons discussed above. Instead, evidence for lack of protein expression has been moved into main Figure 1 (c-f) of our revised manuscript. This now consists of quadruplicate samples for 2 timepoints: it is clear that *DLG2* protein is only expressed in WT cells.

3. *There is a very significant difference in the number of CTIP2 positive cells between KO1 and KO2 at P40, P50 and P60. This indicates that the age at which the cultures reach maturity is a highly variable process. As CTIP2 density (or the neuronal maturity that it represents) appears to underlie the SZ associated differential expression, this presents an issue with claiming the changes are really DLG2 associated: perhaps it's just a fluke relating to how the cultures were handled. From reading the methods, I cannot quite tell if the wild type cultures were handled in exactly the same way as the genetically modified cultures: were the wild type cultures also nucleofected and FACS sorted prior to plating? My impression is that no attempt was made to pseudo-nucleofect the control cultures in the same manner. If this processes stresses the cells then it seems plausible that this could have stressed the cells and delayed their differentiation as a result. If this is the case, would it be possible to try testing whether the process affects the CTIP2 density as shown in SuppFig6? Recognising that getting any experiments done at present, at the very least Supplemental Figure 6 should be included as a main figure, the significance of the difference between the two cultures stated and this possibility clearly noted.*

Response: The reviewer suggests that WT and KO cell-lines may have been treated differently – KO lines being subject to the additional stresses of nucleofection and FACS sorting – and that this has delayed the differentiation of KO cells by altering *CTIP2* expression. As stated in the Methods ('*DLG2 Knockout hESC line generation*', p12), both WT and KO lines were treated identically, going through the same process of nucleofection and FACS sorting:

"H7 hESCs... were nucleofected ..., FACS sorted on the following day and plated at a low density (~70 cells/cm²) for clonal isolation. 19 clonal populations were established with 6 WT and 13 mutant lines after targeted sequencing of the exon 22. One WT and two homozygous knockout lines were chosen for study."

Consequently, in the main text we refer to the WT line as a sister line, rather than a parent line. We appreciate that this distinction may not be obvious enough. In order to clarify, when discussing KO line generation in the main text we now explicitly state that (p3) “All subsequent analyses compared these lines to an isogenic WT sister line that went through the same procedure but remained genetically unaltered”. Differences in WT and KO expression are a consequence of *DLG2* KO, not artefact.

Concerning the differences between KO1 and KO2 lines, we explicitly note this in the main text, referring the reader to Supplementary Figure 6 (Supplementary Figure 8 in revised manuscript). There is a notable degree of variability between experiments, but it does not appear to be the case that expression in one line is consistently recovering more quickly than the other: while the KO2 data shows greater recovery in the percentage of CTIP2 expressing cells, there is greater recovery at the total protein level in KO1 samples (Supplementary Figure 8). Under our hypothesised model of *DLG2* action, greater experimental variability would be expected in KO lines: *DLG2* loss impairs the formation of signal transduction complexes; signalling becomes increasingly stochastic; and the activation of downstream effector pathways regulating neurogenesis (including regulation of transcription) more erratic. Given the variability between KO lines/replicates, we combine the data to obtain a better estimate of the (line-independent) effect of *DLG2* KO.

CTIP2 is a marker of cell-type identity, not maturity, being highly expressed in Layer V excitatory cortical neurons. By day 60, approximately 20% of our cultures are CTIP2⁺ Layer V cells and 30% are earlier born TBR1⁺ Layer VI cells. It should also be noted that high *in vivo* expression of the early-increasing^{-/-} set, which contains CTIP2, occurs later in development than peak expression of the early-transient^{-/-} set (revised Figure 7), as predicted by our *in vitro* analyses (Figure 4). Down-regulation of CTIP2 expression is therefore likely to contribute to *DLG2* KO gene expression changes and cellular phenotypes in less than 50% of neurons, and even here it is extremely unlikely to be the primary driver of *DLG2*-dependent changes.

4. All bar plots used to show differences between groups, e.g. Fig 2c,d,e,f,g,h and Fig3a,b,c,d,f,g should be replaced with boxplots, jitter plots or ideally, estimation plots (<https://www.nature.com/articles/s41592-019-0470-3>).

Response: We appreciate this suggestion from the reviewer, as these formats do indeed give more information to readers; we have updated the plots as requested.

5. No data / code availability sections. The descriptions of some sections of the computational analysis are too vague; sharing the code would clear up any ambiguities. All scripts relating to differential expression etc should just be posted on GitHub or similar. The datasets were not made available to reviewers. How the differential expression was obtained for the single-cell RNA-seq is particularly unclear: did they combine data from all cell types at a particular time point?

Response: The statement on data availability was missing from the Methods, for which we apologise – it has now been added. However, the information was available in the accompanying reporting summary, where it states (top of p2):

“RNAseq data generated by this study have been deposited in the European Nucleotide Archive with the accession number PRJEB35773.”

As the European Nucleotide Archive does not allow pre-release access, we have additionally deposited the data in GEO (accession GSE172199) and an access token has been passed to the journal editors. Data availability (p29-30) now references GEO. Our analyses largely make use of publicly available software packages, as described in detail in the Methods section. The only bespoke script created for this study was that used for the GO annotation over-representation analysis and iterative refinement; this is now available from GitHub as detailed in the revised Methods (p30). The single-cell RNA-seq differential expression analysis has been replaced by the simpler (and more direct) evaluation of the expression of early neurogenic programs in neurodevelopmental cell-types spanning cortical excitatory neurogenesis *in vivo*, using the same human *in vivo* dataset (revised Figure 7); this is fully described in the revised Methods (p19).

Minor Issues:

1. They state that “this suggests that a significant proportion of SZ common variants may contribute to disease via the disruption of neurodevelopmental pathways harbouring a concentration of LoFi genes”. They state this because SZ genes are enriched for LoFi genes, but LoFi genes are broadly important for nervous system function in adulthood not just neurodevelopment, so this is a misleading statement.

Response: Firstly, the quoted text is proposing a hypothesis to be tested, not stating a verified fact. Secondly, the reviewer’s summary does not fully capture our line of thought, which is laid out in the immediately preceding text (Introduction, p2):

“...nearly 50% of genic SNP-based heritability is captured by loss-of-function intolerant (LoFi) genes. Being under extreme selective constraint, LoFi genes are likely to play important developmental roles and they are known to be enriched for rare variants contributing to autism spectrum disorders (ASD) and intellectual disability/severe neurodevelopmental delay (ID/NDD) as well as SZ. This suggests that...”

We present 3 facts: 1) LoFi genes are highly enriched for schizophrenia risk variants 2) LoFi genes are (by definition) under strong selection, impacting development up to sexual maturity and 3) genetic variation in LoFi genes also contributes to congenital/childhood-onset disorders (with which schizophrenia shares a significant overlap in genetic risk). On this basis, it seems reasonable to hypothesise that schizophrenia common risk variants lying within LoFi genes may contribute to disease by disrupting neurodevelopment. Indeed, this hypothesis is borne out by the analyses we subsequently present (Figure 6).

2. Would be useful to include a UMAP / tSNE / PCA / hclust plot of the datasets to get a sense of where the variation is

Response: we now include a cluster plot and heatmap of the expression data in Supplementary Figure 3. Clustering (panel a) was performed for all samples and shows that at all 4 timepoints all replicates from KO1 and KO2 cluster together. It is also worth noting that KO samples from day 20 onwards cluster with WT samples from the preceding timepoint. This is consistent with the delayed activation of transcriptional programs in *DLG2* KO lines. The heatmap (panel b) provides a more detailed view of gene-level expression; low-quality KO2 day 30 samples are excluded from this plot (as they are from all subsequent bioinformatic analyses).

3. They are not clear about exactly how differential expression was calculated. They state that

expression data from the two DLG2^{-/-} lines were pooled: does this mean the model formula was just expression~genotype? Could they also share differential expression results between the two DLG2^{-/-} lines (and if anything is DE whether that is enriched for schiz heritability at any time points).

Response: DESeq2 employs a generalized linear model with a logarithmic link function to compare expression between two groups of samples, using shrinkage estimation of dispersion to model within-group variability (Love et al., Genome Biology 2014). In our case the two groups correspond to KO samples (all KO1 and KO2 replicates for a given timepoint) and WT samples (all WT replicates for the same timepoint). That is what we mean by the KO1 and KO2 data being pooled. As the Methods state (p18):

“Differential gene expression analysis was performed using the DESeq2 package (v1.24.0) (Love et al., 2014) and differentially expressed genes were considered significant if their p value after Bonferroni correct was <0.05...When analysing differential gene expression in *DLG2^{-/-}* relative to WT, samples from KO1 and KO2 lines were combined i.e. for each timepoint a single differential gene expression analysis was performed, comparing expression in KO1 & KO2 samples against wild-type”

Our purpose here is to identify robust changes caused by *DLG2* deficiency, hence the pooling of KO data; it is not clear what an analysis of differential expression between the two KO lines would add. Under a model in which *DLG2* KO leads to stochastic activation of transcriptional programs, it would not be surprising if between-line DEGs captured elements of schizophrenia-associated programs.

4. In SuppFig6d,e,f at P50 for KO1+KO2, CTIP2 expression is 50% of WT... but for KO1 it's ~10% and for KO2 it's ~2%. So how does 10% + 2% get to 50%?

Response: In Supplementary Figure 6, panel 6a was erroneously replicated in place of 6d. This has been replaced with the correct panel in the revised manuscript (now Supplementary Figure 8). We thank the reviewer for pointing this out.

5. They say “captured the signal”. Should be “captured the heritability enrichment” or similar.

Response: Throughout the manuscript we use MAGMA to test whether gene-sets are enriched for common variant association. To talk about a subset of genes capturing the association signal present in a larger set is perfectly standard terminology. At no point do we directly estimate or test for heritability enrichment using methods for partitioning heritability: to use ‘heritability enrichment’ or similar here would be potentially misleading.

6. They say “while complexes regulating postsynaptic information processing have been robustly implicated in SZ, this represents the first evidence that electrical properties underlying information transmission are also disrupted”. It's a bit much to say this on the basis of a GO enrichment.

Response: This statement was not solely based on GO term enrichment, as we hope should be evident from the section in which it appears (*Convergence of genetic risk on perturbed action potential generation* p6) and the accompanying figure panel 4f (Figure 4e in revised manuscript). In this section, GO terms whose genes are enriched in schizophrenia-associated programs are simply the starting point. Taking each such GO term, we test whether genes in the program

belonging to that term have significantly higher GWAS association for schizophrenia than the program as a whole (testing for association enrichment in MAGMA whilst covarying on the program as a whole, then correcting p-values for the overall number of terms tested). The first paragraph in this section reads:

“We next tested whether biological processes over-represented in early-stable^{-/-} or early-increasing^{-/-} (Supplementary Table 5) captured more or less of the SZ association in these programs than expected... Of the 16 subsets for early-increasing^{-/-}, ... *membrane depolarization during action potential* displayed evidence for excess enrichment relative to the program as a whole...”

In summary, i) the early-increasing^{-/-} set is highly enriched for schizophrenia association ii) genes involved in *membrane depolarization during action potential* are over-represented in early-increasing^{-/-} and iii) *membrane depolarization during action potential* genes in early-increasing^{-/-} are more strongly associated with schizophrenia than the early-increasing^{-/-} set as a whole. We then go on to show (see Figure 5) that iv) significant disruption in action potential generation is seen in *DLG2* KO neurons (where the early-increasing^{-/-} set is down-regulated). Our statement is thus supported by far more than GO term enrichment in early-increasing^{-/-}. If there are elements of this analysis that the reviewer feels need to be emphasised more clearly, we would be happy to do so.

7. They say “In summary, *DLG2*^{-/-} neurons have a reduced ability to fire APs and produce less mature APs.” This should be tempered with “at day 50” and should acknowledged that it’s likely to just be delayed rather than a permanent deficiency.

Response: We agree that it is unclear whether this phenotype persists in mature neurons, indeed this applies to all of the phenotypes we observe. We have therefore made a minor modification to this sentence, which now reads:

“In summary, developing *DLG2*^{-/-} neurons have a reduced ability to fire APs and produce less mature APs.”

In the Discussion (end of paragraph 2, p10) we now note that:

“...in future studies it will be important to investigate the persistence of these phenotypes ...”

8. A supplementary table should be included with the differential expression results from the single cell analysis

Response: As noted above (response to 9) the differential expression analysis of single-cell data has been removed from the revised manuscript, being replaced by a direct evaluation of early neurogenic program expression in the *in vivo* single cell RNA-seq data.

9. They note in the methods that the day 30 KO2 samples were dropped: could they be a bit more explicit that this meant that only KO1 samples were used at day 30.

Response: We have modified the text in the Methods section (p18), which now reads:

“... This revealed a high level of duplicate reads in day 30 KO2 samples (~72% compared to an average of 23% for other samples). These samples were removed prior to further analyses, which were thus performed on KO1 and WT samples for this timepoint.”

10. On page 15 they report for day 30 the correlation between WT vs KO, performed for each cell line separately; it's remarkably high. Could they report the same values for other days?

Response: To assess the impact of sample dropout for day 30, we reported the correlation between KO1 v WT and KO2 v WT for day 20 samples as this was the timepoint nearest to day 30, also lying close to the onset of neurogenesis, and which had the most comparable number of differentially expressed genes. We agree that it would be more transparent to present data for other days as well. In addition, the new data presented in Supplementary Figure 3 and revised Figure 7 also indicate that sample dropout does not significantly impact our analyses. We have therefore revised this section of the text (p18-19), which now reads:

“To assess the impact of sample dropout at day 30, we investigated the similarity in gene expression between lines by clustering all KO1, KO2 and WT samples (Supplementary Figure 3a). At all 4 timepoints, all replicates from KO1 and KO2 cluster together: while KO2 samples from day 30 are not of sufficient quality to be used with confidence in further analyses, they are clearly similar to KO1 day 30 samples. We also performed differential expression analyses separately for each line (i.e. KO1 v WT and KO2 v WT) at all other timepoints. The overlap in expressed genes accounted for >98% of the genes expressed in each line and gene expression fold change was highly correlated between KO1 v WT and KO2 v WT (Spearman's $\rho^{\text{day } 15} = 0.67$, $\rho^{\text{day } 20} = 0.95$, $\rho^{\text{day } 60} = 0.75$). Over-representation odds ratios for GO terms also remain well correlated for significantly up-regulated ($\rho^{\text{day } 15} = 0.70$, $\rho^{\text{day } 20} = 0.92$, $\rho^{\text{day } 60} = 0.67$) and down-regulated regulated ($\rho^{\text{day } 15} = 0.55$, $\rho^{\text{day } 20} = 0.95$, $\rho^{\text{day } 60} = 0.56$) genes. We noted that agreement between lines was greatest for day 20, which also lies close to the onset of neurogenesis and displays a high level of differential expression between KO and WT lines, comparable to that for day 30 (Figure 1g). Further indicating a limited impact for sample dropout, phenotypes predicted by GO term analysis of differential expression at day 30 (deficits in neuron migration, morphology and action potential generation) were experimentally validated (Fig. 3 & 5); and all early neurogenic transcriptional programs identified in these data were shown to possess an identical profile of expression across human neurodevelopmental cell-types *in vivo* (Fig. 7).”

Reviewer #3

The manuscript by Sanders et al. investigates the neurodevelopmental role of a synaptic protein, DLG2, during directed differentiation of human pluripotent stem cells towards the dorsal telencephalon. DLG2 has been implicated in the etiology of Schizophrenia and the authors demonstrate that loss of DLG2 function alters a neurogenic transcriptional program that is similarly found in other neuropsychiatric/neurodevelopmental diseases suggesting this transcriptional program is a point of convergence. Specifically, the authors uncovered a cell type specific delay in excitatory deep layer neuron generation (marked by BCL11B/CTIP2 expression) during cortical neurogenesis. DLG2 knockout neurons were shown to have decreased migration, neuronal complexity and functionality. Overall this is a well-presented manuscript and provides evidence linking DLG2 as a major factor of convergence in neurodevelopmental disorders. I do have some concerns on the regionalization and cell type specificity from the differentiations

performed and how it might impact the interpretation and the overall conclusions of the data summarized below:

1. The authors identify the day 30 timepoint (start of neurogenesis) on their differentiation where a particular enrichment for schizophrenia related genes deviated from wildtype controls (Figure 2B and 4B) and surprisingly is not enriched in the late stage (day 60). This is a very interesting point since the culture should be enriched for immature neurons and neural progenitors and *DLG2* may be involved in other processes in addition to the synapse. However, looking at the gene expression changes in Supplemental Table 3, it appears that several genes such as *DLX1*, *DLX2*, *GBX2*, *NKX2.1* and *FOXP1* are significantly and differentially expressed at this particular timepoint between WT and mutant.

It would be important for the authors to investigate and confirm the cell type specificity of the cells generated other than simply using *TBR1*, *CTIP2* and *SATB2* as markers for cortical excitatory neurons. The expression of the *DLX*, *GBX* and *NKX* genes suggests that its highly probable that the cells at this particular timepoint were ventralized. *TBR1* can be expressed in the ventral thalamus, *CTIP2* in the striatum and *SATB2* has been described as expressed in the hypothalamus. Excluding these brain regions would greatly strengthen the conclusions of the author's comparative analysis..

Response: As shown in Supplementary Figure 5b (6b in revised), the vast majority of cells in our cultures commit to a dorsal telencephalic fate long before neurogenesis, with over 90% of day 20 cells expressing key markers of dorsal telencephalic identity (*PAX6*⁺ and *FOXP1*⁺). Once progenitors have committed to a dorsal fate, they do not produce ventral-derived neurons unless treated with region-specific inductive cues such as *SHH* (Hitoshi et al., Development 2002). In line with this, either *SHH* or *Activin A* are essential to differentiate hPSCs into ventral progenitors and their derivatives such as cortical interneurons (Maroof et al., Cell Stem Cell 2013; Nicholas et al., Cell Stem Cell 2013; Cambray et al., Nat Comms 2012) and striatal projection neurons (Arber et al., Development 2015). Without these exogenous inductive molecules in our culture, we do not expect to produce any ventral-derived neurons from the >90% dorsal committed progenitors. Furthermore, examining human foetal brain single-cell RNAseq expression data (Nowakowski et al., Science 2017) we find that *DLX1* and (to a greater extent) *DLX2* are expressed in sub-populations of forebrain excitatory neurons *in vivo*. We therefore expect well over 90% of cells in our cultures to develop into cortical excitatory neurons and that – if present – ventral sub-types constitute a very small proportion. To confirm this, we have performed extensive extra staining (revised Supplementary Figure 6). These additional analyses confirm the overwhelmingly dorsal telencephalic identity of our cultures, with at most 1% of cells expressing ventral genes *GBX2*, *NKX2.1* and *OLIG3* at days 20 and 30 in WT and KO cultures (Supplementary Figure 6c-d: data also presented for *DLX1*, although as noted this is likely to correspond to a mix of both dorsal and ventral cells).

In addition, we have also investigated the presence of thalamic, striatal and hypothalamic neurons as suggested by the reviewer. Our day 30 cultures contain no striatal (*GABA*⁺/*CTIP2*⁺/*FOXP1*⁺/*DARPP32*⁺) neurons; no hypothalamic (*SATB2*⁺/*FOXP1*⁻) neurons; and a very small proportion (~2% of *TBR1*⁺ cells, which corresponds to <0.4% of all cells) of thalamic (*TBR1*⁺/*GBX2*⁺) neurons (Supplementary Figure 6e-g). The data reveals a significant decrease in the number of *DLX1*⁺ and *NKX2.1*⁺ cells in KO cultures at day 30 (Supplementary Figure 6d), consistent with bulk RNA-seq differential expression. These data indicate that bulk RNA-seq analysis is capable of picking up changes in genes uniquely expressed in a very small proportion

of cells (1% or less). Beyond a small number of marker genes unique to these cells, bulk RNA-seq reliably captures expression in the dorsal telencephalic population that comprises the vast majority (>90%) of cells. We note that the 3 transcriptional programs which we derive from bulk RNA-seq data are clearly expressed in the expected developmental sequence during cortical excitatory neurogenesis *in vivo* (revised Figure 7). In summary, we confirm that our cultures overwhelmingly produce dorsal forebrain cortical excitatory neurons and that other cell-types do not significantly impact our results.

2. Somewhat related to the point above, neural progenitors in culture typically generate the cortical layers in a sequential and time dependent fashion. The data the authors present in Figure 1F show a modest increase in TBR1+ cells over time suggesting that the majority of these neurons are born on day 30. Subsequently, there appears to be no change in the generation of SATB2+ cells in the same time course (Supplemental Figure 5H) suggesting that those neurons are either not born yet (based on the low percentage of cells), or already established at day 30. The authors need to show supporting evidence that the expression of SATB2 in their cultures indeed represent the cell type they intend. Or they will need to modify their conclusion: “A similar analysis of layer markers TBR1, CTIP2 and SATB2 revealed a significant decrease in CTIP2+ cells but a comparable proportion of TBR1+ and SATB2+ neurons for all timepoints investigated (Fig. 2d, e, g-i, Supplementary Fig. 5).”

Response: Our discussion of SATB2 expression was not as clear as it should have been, for which we apologise. *In vitro* differentiation mimics *in vivo* neurodevelopment, with deep layer cortical neurons generated prior to upper layers (van de Leemput et al., Neuron 2014; Varrault et al., Stem Cells Dev 2019): at this early stage of 2D differentiation (days 30-60), we do not expect to see the generation of upper layer neurons. Although highly expressed in upper layer neurons, SATB2 is also expressed for a considerable length of time in developing CTIP2+ deep layer cortical neurons in human foetal brain, particularly in prefrontal cortex (Nowakowski et al., Science, 2017). Therefore, SATB2 expression in our cultures is most likely to arise from deep layer CTIP2+ neurons. To show this explicitly we have performed additional staining; this reveals that 100% of our SATB2+ cells co-express CTIP2 (revised Supplementary Figure 7f), proving they are deep layer neurons rather than later born upper layer neurons. We have incorporated these data into the results (p4) and modified the text to make it clear that SATB2 is not labelling upper layer neurons but is expressed within CTIP2+ deep layer neurons.

3. The data that indicates a delay in the generation of CTIP2+ neurons due to the loss of DLG2 needs additional experiments to support this conclusion. The authors show that the generation of TBR1 and SATB2 is roughly equivalent between WT and KO, but CTIP2+ neurons are dramatically lost/delayed in the cells derived from the KO line. Eventually, CTIP2+ cells “recover” with time in the KO line. Are the CTIP2+ cells actually coming from cortical patterned NSCs? Lineage tracing or clonal labeling of the starting radial glial cells will confirm that the DLG2 KO radial glia will produce the cortex in an “out of order” manner. Alternatively, sorting out the cortical radial glial cells from these timepoints and triggering the differentiation would help to address this question as well.

Response: As noted in our response to the previous comment, SATB2 is not labelling upper layer neurons; we would not conclude that CTIP2+ cells are being born ‘out of order’ and have revised the text to clarify this point. Concerning the origin of CTIP2+ cells, our cultures do not contain a substantial proportion of non-cortical NSCs: over 90% of our progenitors commit to a dorsal telencephalic fate; there are no more than 1% ventral cells in either WT or KO cultures;

and CTIP2⁺ cells do not express any of the striatal markers suggested by the reviewer (see response to Reviewer #3 comment 1). Given these data it is clear that the CTIP2⁺ population is not derived from non-cortical radial glia or generated 'out of order' – lineage tracing/clonal labelling is not required to prove this.

4. In both the morphology and the electrophysiology studies, it is unclear how the authors decided which neurons to assay. Given the heterogeneity of the neurons in the dish at day 50 (Ephys) and d70 (morphology), and the potential delay in neurogenesis of CTIP2⁺ neurons from the DLG2 KO, could some of the neurons be at different stages in development? How did the authors decide on the criteria used to distinguish the mature and immature neurons rather than an arbitrary timepoint? More details of this would be beneficial towards the interpretation of their data.

Response: For the morphology analysis, as stated in the Method (p22), random fields were imaged for all samples regardless of genotype. For electrophysiology experiments, the most morphologically mature neurons were patched in each culture; hence the most comparable subpopulation of cells from each genotype was compared. This information has now been added to the Methods (p22).

Reviewers' Comments:

Reviewer #2:

Remarks to the Author:

I remain unconvinced that their early-increasing or late transcriptional programs represent more than a transition towards neuronal genes and do not think they have really attempted to show otherwise. Their counter-argument is that they have performed MAGMA using condition-residualize against all the genes expressed in any of the wildtype / knockout cultures. But this set of genes must include almost the whole human transcriptome. Firstly, they should be clearer about how many genes this set includes. But if they really want to show that these are transcriptional programs that really represent something other than a transition to 'mature neurons' then they should perform condition-residualize in a manner that accounts for the fact that highly expressed neuronal genes are more associated with schizophrenia than weakly expressed genes. Another way they could attempt to show this is perhaps by comparison against data from adult neurons, in a similar manner to their current scRNA-seq study: if these are really 'developmental programmes' then the early-increasing and late genes should comprise some genes which are not highly expressed in mature neurons, and that set of genes should still be enriched for disease heritability. It would be simpler to just phrase things in a manner which allows for this and which doesn't suggest they have been strict in controlling for neuronal background. I certainly don't think that they've shown that "the burden of proof lies in showing that their function in mature neurons also contributes to disease".

The wording of the article has been improved and some claims are now made in more measured tones, but jumps of logic remain. They do still include sections such as "Based on its known function, and involvement of invertebrate Dlg in the developmental Scrib signalling module we propose that DLG2 links cell-surface receptors to signal transduction pathways regulating the activation of neurogenic programs (Supplementary Fig. 9). We hypothesise that stochastic signalling in DLG2^{-/-} lines delays and impairs transcriptional activation". These imply that DLG2 is directly involved in transcriptional machinery which their data does not pertain to.

They suggest that it is not a problem that their differential expression results don't show that Dlg2 is down-regulated, as such, they did not agree to report this in the paper. My concern was not however that the cell lines might not have had the gene knocked down, rather I am worried that their differential expression results are not well powered / controlled. While they are correct that transcript expression level of knockout genes is not always well detected, in my experience, it still is downregulated in most cases. I say, let the reader decide for themselves whether they think this is important.

They suggest that 'greater experimental variability would be expected in KO lines'. What is seen though is not greater variability within the knockout lines, the issue is substantial variability between the knockout lines. There does in fact seem to be little variance within KO2, it just doesn't resemble KO1 in any way based on Supp Fig 8.

They say that they have gotten rid of the bar plots. However, the great majority of graphs in the paper remain bar plots. The bar plots should be removed as previously requested.

They have now shared some bespoke code but do not provide markdown files or notebooks to enable analyses to be reproduced, instead noting that "our analyses largely make use of publicly available software packages" (not something that inhibits the use of notebooks/markdown/scripts).

They have kept the text stating: "LoFi genes are likely to play important developmental roles and are known to be enriched for rare variants contributing to autism spectrum disorders (ASD) and intellectual disability/severe neurodevelopmental delay (ID/NDD) as well as SZ. This suggests that a significant proportion of SZ common variants may contribute to disease via the disruption of neurodevelopmental pathways harbouring a concentration of LoFi genes". In stating this they ignore that intellectual disability is generally caused by loss of function mutations, while schizophrenia is associated with common variation, with the former being more likely to show severe developmental effects. I would argue that loss of function of important neuronal gene alters

neurodevelopment, but common variants associated with schizophrenia in the same genes don't affect neurodevelopment in the same way and hence cause disease in adults. I just don't see that the argument follows that because schizophrenia is associated with LoFi genes that schizophrenia genes must act via neurodevelopment. Perhaps the wording could be moderated?

I requested that they also test for differential expression between the two knockouts, and test whether this is associated with schizophrenia heritability. My interest herein is about how specific the differential expression / MAGMA enrichments they've seen are: it is a quick control experiment to establish that. They rejected to perform this analysis.

Reviewer #3:

Remarks to the Author:

The revised manuscript of Sanders et al have made considerable improvements and have clarified the concerns I had with the manuscript. I wish them the best in their future studies.

Reviewer #4:

Remarks to the Author:

The response to the first round of reviews is satisfactory. Overall, it is suitable for publication in Nature Communications.

We are pleased to note that Reviewers #3 and #4 are completely satisfied with our response and we thank Reviewer #3 for their kind words. We now respond to the remaining comments from Reviewer #2, performing substantial additional work that not only strengthens the conclusions of the study but also provides novel insight into the relationship between developmental and mature neuronal pathways contributing to schizophrenia. We include the full text of their response but have split the first paragraph of remarks into two sections as they raise distinct points. We note that Reviewer #2 did not address a number of the points raised in our previous response: these have been reiterated where appropriate.

Reviewer #2 (Remarks to the Author):

1. I remain unconvinced that their early-increasing or late transcriptional programs represent more than a transition towards neuronal genes and do not think they have really attempted to show otherwise. Their counter-argument is that they have performed MAGMA using condition-residualize against all the genes expressed in any of the wildtype / knockout cultures. But this set of genes must include almost the whole human transcriptome. Firstly, they should be clearer about how many genes this set includes.

The all^{WT+KO} set that we condition on contains 14,274 genes; this is comparable to the 14,134 expressed in mature CA1 pyramidal neurons (genes with specificity score > 0) according to (Skene et al., 2018), to which we will return later. We would note that if all^{WT+KO} had contained the entire transcriptome, then it would show no enrichment for SZ association since we are performing a competitive test: in actual fact it is highly enriched ($P = 8.03 \times 10^{-21}$). In addition to pointing out that we condition on all^{WT+KO} , we raised two further points in our previous response:

- i) Genes up-regulated on day 30 in wild-type cultures that are not down-regulated on *DLG2* KO are not enriched for association.
- ii) Genes only up-regulated after day 30 (the 'late' set, which the reviewer mentions above) are also not enriched for association.

If we were only picking up a generic 'neuronal' signal captured by genes up-regulated during early neurogenesis, then the gene-sets outlined in i) and ii) should be enriched for association. The reviewer has not addressed either of these points. In our revised manuscript we include two additional analyses:

- iii) Rather than using a single set of expressed genes to cover all timepoints, we test for association in genes up-/down-regulated at a given timepoint whilst conditioning on genes expressed specifically at that timepoint. This makes no difference to our results (p4, Supplementary Table 4).
- iv) Since our data comes from bulk RNAseq, the time-point specific expressed genes used for iii) will still contain a mix of progenitor and neuronal genes. We therefore identified genes expressed in new-born and developing cortical excitatory neurons *in vivo* (see p6 of revised manuscript; data from Nowakowski et al., 2017). Early-increasing^{-/-} and early-stable^{-/-} are still highly enriched for SZ GWAS association when conditioning on these genes (see Supplementary Table 6, where we report analyses using both stringent and relaxed criteria to identify *in vivo* expressed genes). Furthermore, all^{WT+KO} displays significant association that is not captured by *in vivo* new-born/young neuron-expressed genes.

The above analyses conclusively show that SZ association in early-increasing^{-/-} and early-stable^{-/-} is not due to a general enrichment for signal in neuronal/neuronal-lineage genes. We

continue to condition our analyses on all^{WT+KO} as this best captures the general SZ signal in neuronal-lineage genes present in our data (see iv above).

2. But if they really want to show that these are transcriptional programs that really represent something other than a transition to 'mature neurons' then they should perform condition-residualize in a manner that accounts for the fact that highly expressed neuronal genes are more associated with schizophrenia than weakly expressed genes. Another way they could attempt to show this is perhaps by comparison against data from adult neurons, in a similar manner to their current scRNA-seq study: if these are really 'developmental programmes' then the early-increasing and late genes should comprise some genes which are not highly expressed in mature neurons, and that set of genes should still be enriched for disease heritability. It would be simpler to just phrase things in a manner which allows for this and which doesn't suggest they have been strict in controlling for neuronal background. I certainly don't think that they've shown that "the burden of proof lies in showing that their function in mature neurons also contributes to disease".

The reviewer appears to be conflating what we would consider to be two quite different types of analysis. The first is when we wish to test association enrichment in a gene-set A which is part of a larger set B that is itself enriched for association (e.g. neurogenic programs as subsets of neuronal-expressed genes). To show that risk variants specifically implicate the function of A (i.e. A is significantly more associated with disease than would be expected for a random set of genes from B) we need to test A for genetic enrichment while conditioning on B: if there is no association then we can conclude that the disease-relevant biology underlying association in B is not specifically concentrated in A. This is the type of analysis discussed in our response to 1 above, where we show that SZ genetic risk is highly concentrated in neurogenic programs and is not simply a consequence of the genes in these programs being neuronal/neuronal-lineage expressed (i.e. no different to a random collection of neuronal genes).

The second type of test is when we compare the association signal between two sets that may overlap but where one is not a direct sub-component of the other e.g. genes expressed in cell-type A and cell-type B. In this case we cannot make strong inferences about which is disease relevant. For example, if association enrichment in cell-types A and B overlaps completely (i.e. no association remains when conditioning A on B and vice versa) then we cannot say whether the signal 'belongs' to cell-type A or B (or both). It makes no difference whether association has previously been reported for one of them: if we randomly chose one cell-type to test first, then later tested the other and performed conditional analyses, we could not conclude that the first cell-type is the disease-relevant one. Even if cell-type A remains significant when conditioning on B, but B does not when conditioning on A, then all we can say is that the most parsimonious explanation is that cell-type A is the disease-relevant one – we cannot rule out the involvement of cell-type B.

The comparison suggested by the reviewer belongs to the second type of analysis: by itself, testing the overlap between neurodevelopmental and mature neuronal pathways does not tell us which is disease-relevant. It is not valid to claim that any overlapping signal should be interpreted as supporting the disease relevance of mature pathways alone, not only due to the technical limitations of this type of analysis but also on biological grounds: disruption of neurodevelopment is known to contribute to SZ and there is no evidence that risk variants in genes expressed during early neurogenesis have no effect until neurons mature. On the contrary, we show that common and rare variation in neurogenic pathways contribute to early-onset disorders ASD and ADHD (Figure 6 c-f), both of which share significant common variant heritability with SZ. It is extremely unlikely that SZ-associated SNPs concentrated in neurogenic

programs have no effect until adulthood when a substantial proportion of those very same SNPs also confer risk for ASD and ADHD.

In addition to the points raised above, we have also performed analyses along the lines requested. SZ GWAS association has been reported in genes with relatively high expression in mature CA1 pyramidal neurons compared to other brain cell-types (Skene et al., 2018; PGC Schizophrenia Working Group, medRxiv 2020). Taking the top 10% of genes with highest CA1 pyramidal neuron specificity score (pyramidal^{high}), we now show (p9) that pyramidal^{high} genes significantly overlap early-stable^{-/-}, early-increasing^{-/-} and late sets (Fig. 7a). This overlap captures GWAS association in pyramidal^{high} (non-overlapping genes $P_{\text{corrected}} = 0.052$) but not early-stable^{-/-} or early-increasing^{-/-} (non-overlapping genes $P_{\text{corrected}} = 9.87 \times 10^{-6}$ and 0.0015 respectively) (Fig. 7b). In line with our comments above, the most parsimonious explanation is that SZ GWAS association primarily reflects perturbation of early development. However, we think it is more likely that disruption of neurodevelopment and mature neuronal function both contribute to disease (see Discussion, p11-12).

3. The wording of the article has been improved and some claims are now made in more measured tones, but jumps of logic remain. They do still include sections such as "Based on its known function, and involvement of invertebrate Dlg in the developmental Scrib signalling module we propose that DLG2 links cell-surface receptors to signal transduction pathways regulating the activation of neurogenic programs (Supplementary Fig. 9). We hypothesise that stochastic signalling in DLG2^{-/-} lines delays and impairs transcriptional activation". These imply that DLG2 is directly involved in transcriptional machinery which their data does not pertain to.

The text quoted by the reviewer does not involve a jump in logic. In mature neurons, DLG2 is well known to link cell-surface NMDA receptors to downstream signal transduction pathways regulating gene expression (as well as other processes). We are simply proposing that DLG2 may play a similar role during early development. At no point do we claim to have shown this, as this would indeed be jumping to unjustified conclusions. We are stating a valid, entirely reasonable hypothesis for the mode of action of DLG2 based on its known functional properties as a scaffold protein and its known role in mature neurons. At no point do we claim that "*DLG2 is directly involved in transcriptional machinery*", as is evident from our proposed model above and Supplementary Figure 9.

4. They suggest that it is not a problem that their differential expression results don't show that Dlg2 is down-regulated, as such, they did not agree to report this in the paper. My concern was not however that the cell lines might not have had the gene knocked down, rather I am worried that their differential expression results are not well powered / controlled. While they are correct that transcript expression level of knockout genes is not always well detected, in my experience, it still is downregulated in most cases. I say, let the reader decide for themselves whether they think this is important.

The reviewer accepts that there is no requirement for us to see a decrease in *DLG2* mRNA in knockout lines: in other words, *DLG2* mRNA abundance tells us nothing about the quality of our data. Furthermore, we note that the main text already directs readers to analyses showing a decrease in *DLG2* mRNA at the exon level for days 30 and 60 (p3, Supplementary Figure 4); this point was raised in our previous response but has not been acknowledged by the reviewer. It is therefore not valid to draw into question our differential expression analysis based on *DLG2* mRNA levels. The main text and figures of a study report the robust, primary findings: gene-level mRNA data from which no firm conclusions can be drawn does not warrant inclusion and simply risks misleading readers. Consequently, we see no valid reason for insisting that we

report *DLG2* gene-level mRNA expression in the main body of the paper; differential expression statistics for all genes can be found in Supplementary Table 3.

5. *They suggest that 'greater experimental variability would be expected in KO lines'. What is seen though is not greater variability within the knockout lines, the issue is substantial variability between the knockout lines. There does in fact seem to be little variance within KO2, it just doesn't resemble KO1 in any way based on Supp Fig 8.*

It is not valid to suggest that KO1 does not resemble KO2 “in any way” based on a small sample of immunocytochemistry (ICC) and Western blot data-points for a single gene. We expect to see variation both within and between lines and note that expression in KO2 lines is not consistently less variable than that in KO1, even within Supplementary Figure 8 (ICC day 30, panels b & c). We also note that ICC expression for other genes is clearly comparable between KO1 and KO2 (Supplementary Figure 5e). The overall pattern of gene expression in each line is a far more reliable indicator of the relationship between them: based on this, KO1 and KO2 closely resemble each other and are distinct from WT (Supplementary Figure 3).

6. *They say that they have gotten rid of the bar plots. However, the great majority of graphs in the paper remain bar plots. The bar plots should be removed as previously requested.*

In the previous revised figures, we added dot plots on top of the bar plots to show the dispersion and density of the data. Now, in line with the journal's formatting instruction, we have changed them either to dot plots or violin plots depending on the number of datapoints. We also provide source data for all graphs presented in the study.

7. *They have now shared some bespoke code but do not provide markdown files or notebooks to enable analyses to be reproduced, instead noting that "our analyses largely make use of publicly available software packages" (not something that inhibits the use of notebooks/markdown/scripts).*

We have collated scripts for the RNAseq and bioinformatic/human genetic analyses and make them available with the manuscript.

8. *They have kept the text stating: "LoFi genes are likely to play important developmental roles and are known to be enriched for rare variants contributing to autism spectrum disorders (ASD) and intellectual disability/severe neurodevelopmental delay (ID/NDD) as well as SZ. This suggests that a significant proportion of SZ common variants may contribute to disease via the disruption of neurodevelopmental pathways harbouring a concentration of LoFi genes". In stating this they ignore that intellectual disability is generally caused by loss of function mutations, while schizophrenia is associated with common variation, with the former being more likely to show severe developmental effects. I would argue that loss of function of important neuronal gene alters neurodevelopment, but common variants associated with schizophrenia in the same genes don't affect neurodevelopment in the same way and hence cause disease in adults. I just don't see that the argument follows that because schizophrenia is associated with LoFi genes that schizophrenia genes must act via neurodevelopment. Perhaps the wording could be moderated?*

Whether the reviewer agrees with it or not, this text simply lays out the authors' reasoning at the start of the study. The reviewer's argument concerning loss-of-function (LoF) mutations and common variation overlooks a number of key points:

- i) LoF mutations also contribute to SZ risk
- ii) Furthermore, we explicitly show that the early-stable^{-/-} set is enriched for *de novo* LoF mutations found in SZ patients, with association restricted to LoFi genes (Fig. 6 c-d).
- iii) SZ shares substantial common variant heritability with early-onset disorders ASD and ADHD.
- iv) Again, we explicitly show that early neurogenic programs are enriched for common variant association for ASD and ADHD, with association restricted to LoFi genes (Fig. 6 e-f).

Clearly, both rare and common variation in neurogenic programs (and specifically within LoFi genes belonging to these programs) contributes to developmental deficits that manifest early in life. We note that mature neuronal (pyramidal^{high}, see response to 2) genes that do not overlap neurogenic programs are not enriched for LoFi genes: if anything they are slightly depleted, although not significantly so (OR = 0.95, P = 0.63). The text is already moderately worded as the authors' working hypothesis rather than a statement of fact.

9. I requested that they also test for differential expression between the two knockouts, and test whether this is associated with schizophrenia heritability. My interest herein is about how specific the differential expression / MAGMA enrichments they've seen are: it is a quick control experiment to establish that. They rejected to perform this analysis.

In our previous response we pointed out that this was not an informative 'control' analysis. At a number of places in our response we also drew attention to the fact that the neurogenic programs we uncover are also identifiable in independent scRNAseq data for *in vivo* neurodevelopmental cell-types (Figure 7), confirming the biological validity of our expression analyses. The reviewer has not responded to these points. Nevertheless, we have performed the requested analysis: genes differentially expressed between knockout lines are not enriched for SZ association at day 15 ($N_{\text{gene}} = 734$; $P = 0.51$), day 20 ($N_{\text{gene}} = 210$; $P = 0.16$) or day 30 ($N_{\text{gene}} = 1079$; $P = 0.17$); no genes were significantly differentially expressed between the two lines at day 60.

Reviewer #3 (Remarks to the Author):

The revised manuscript of Sanders et al have made considerable improvements and have clarified the concerns I had with the manuscript. I wish them the best in their future studies.

Reviewer #4 (Remarks to the Author):

The response to the first round of reviews is satisfactory. Overall, it is suitable for publication in Nature Communications.

Reviewers' Comments:

Reviewer #3:

Remarks to the Author:

This review is an attempt to determine whether the original reviewer #2's comments were sufficiently addressed by the authors. I believe that the majority of the author's rebuttal are reasonable to without asking for more experimental or analytical revisions. However, I think the problem may be due to the clarity of the manuscript.

Major concern: The sentence structures are hard to follow and sometimes feels like a lengthy outline. For example, in the introduction "Study design is outlined in Fig. 1a.". Similarly, the sentences jump between synaptic regulation, neurodevelopment, common and rare variants all leading to DLG2. This is difficult to follow as well as other sections in the manuscript.

Minor concerns:

1. In point #3 – The authors can resolve this point of contention by placing a 'could' in the hypothesis. It should be clear that many pathways converge on affecting gene expression networks and DLG2 should not be excluded, but perhaps not as direct as the reviewer interpreted.
2. In point #4 – The authors should perhaps state that the mutations they generate could cause non-sense mediated decay or result in a poly-A mRNA. From their Supplementary Figure 1 the knockouts should be frameshifted so the results could be different if poly A RNA was not enriched and if the authors isolated total RNA like they did for the RNA sequencing, perhaps that is why there is no difference in DLG2 RNA levels. To resolve this is just to state this clearly.
3. In point #5 – The reviewer is pointing to the variation of CTIP2 expression between KO1 and KO2. This experiment performed by the authors did not show similar trends between the two knockout lines and I believe is a valid criticism. This is likely due to the rate of cortical differentiation where cells are difficult to control after the initial induction. Without being tested, I think statement made by the authors should be okay.
4. In point #8 – Unless the authors can really demonstrate that SZ is found in fetal brains the comment about neurodevelopmental issues should reflect that unknown. I agree that many shared genes also contribute to risk of ASD and SZ, but even then, it is still a hypothesis that needs testing.
5. In point #9 – "no genes were significantly differentially expressed between the two lines at day 60." This is a bit surprising given the confusion from supplementary figure 8 where some samples did not express CTIP2. Even examining supplementary figure 3, the heatmaps show some differences between KO1 and KO2. Unless this sentence is in relation to the genes associated with SZ?

We are pleased to note that Reviewers #3 is satisfied with the most of our response to Reviewer #2. We now respond to the comments from Reviewer #3, simplifying possibly complex sentences and editing the identified sentences by the reviewer according to their suggestion.

REVIEWERS' COMMENTS

Reviewer #3 (Remarks to the Author):

This review is an attempt to determine whether the original reviewer #2's comments were sufficiently addressed by the authors. I believe that the majority of the author's rebuttal are reasonable to without asking for more experimental or analytical revisions. However, I think the problem may be due to the clarity of the manuscript.

Major concern: The sentence structures are hard to follow and sometimes feels like a lengthy outline. For example, in the introduction "Study design is outlined in Fig. 1a.". Similarly, the sentences jump between synaptic regulation, neurodevelopment, common and rare variants all leading to DLG2. This is difficult to follow as well as other sections in the manuscript.

Response: We have revised the text, simplifying sentence structure in a number of places. In addition to the specific points highlighted by the reviewer, we have also sought feedback from colleagues to identify other parts of the text that may be unclear. A certain amount of jumping around is unfortunately inevitable in the introduction: in order to motivate the study of DLG2 we need to discuss the genetic evidence linking it to disease, experimental evidence that it is expressed early in brain development, and information on its known/hypothesised function. We also need to discuss the relationship between schizophrenia and other disorders and the role of loss-of-function intolerant genes in disease, as these are topics to which we return in the analysis. Modifications are highlighted in the revised text. We hope these changes aid the clarity of the study.

Minor concerns:

1. In point #3 – The authors can resolve this point of contention by placing a 'could' in the hypothesis. It should be clear that many pathways converge on affecting gene expression networks and DLG2 should not be excluded, but perhaps not as direct as the reviewer interpreted.

Response: In line with the reviewer's suggestion, this section of the discussion (p12) now reads:

"Based on its known function and the involvement of invertebrate *Dlg* in the developmental *Scrib* signalling module²⁵, DLG2 **may** link cell-surface receptors to signal transduction pathways regulating the activation of neurogenic programs (Supplementary Fig. 9). We hypothesise that stochastic signalling in *DLG2*^{-/-} lines **due to impaired complex formation could** delay and impair transcriptional activation..."

2. In point #4 – The authors should perhaps state that the mutations they generate could cause non-sense mediated decay or result in a poly-A mRNA. From their Supplementary Figure 1 the

knockouts should be frameshifted so the results could be different if poly A RNA was not enriched and if the authors isolated total RNA like they did for the RNA sequencing, perhaps that is why there is no difference in DLG2 RNA levels. To resolve this is just to state this clearly.

Response: In point#4, reviewer#3 was referring to gene-level *DLG2* RNA expression derived from our RNA sequencing data. Both exon and transcript level analysis of the same data reveal significant depletion of all long coding transcripts in KO lines, most probably indicating nonsense mediated decay as suggested by the reviewer above. We have added text stating that frameshift mutations and premature stop codons were generated in both alleles of KO lines (p3, paragraph 2). This is in addition to the existing reference to exon and transcript analyses indicating degradation via nonsense-mediated decay (p3, paragraph 3). We hope this makes the content clear to the readers.

3. In point #5 – The reviewer is pointing to the variation of CTIP2 expression between KO1 and KO2. This experiment performed by the authors did not show similar trends between the two knockout lines and I believe is a valid criticism. This is likely due to the rate of cortical differentiation where cells are difficult to control after the initial induction. Without being tested, I think statement made by the authors should be okay.

Response: We agree there are differences in CTIP2 expression (we explicitly note this in the main text, p4 last paragraph) which may be due to variability in the rate of differentiation. The reviewer agrees with our statement and no further changes are necessary here.

4. In point #8 – Unless the authors can really demonstrate that SZ is found in fetal brains the comment about neurodevelopmental issues should reflect that unknown. I agree that many shared genes also contribute to risk of ASD and SZ, but even then, it is still a hypothesis that needs testing.

Response: Schizophrenia is well known to have post-natal onset, typically first manifesting between mid-teens and mid-thirties (Solmi et al., 2021). The section of text in question (p2 first paragraph) does indeed frame the developmental contribution of SNPs as a hypothesis, not a fact:

“We hypothesised that a significant proportion of SZ common variants may contribute to disease via the disruption of neurodevelopmental pathways...”

Furthermore, in the discussion we note the post-natal onset of schizophrenia and explicitly state our interpretation of the common variant analyses as a hypothesis:

“...SZ onset extends from late childhood well into adulthood⁶¹. We hypothesise that vulnerability to SZ is primarily established during early neurodevelopment, and that this is subsequently compounded by a gradual accumulation of deficits during circuit maturation due to both external stressors and the impaired function of neurogenic pathways that remain operant throughout childhood and into adulthood.”

The text is therefore in line with the reviewer’s comments above and no further changes are necessary.

5. In point #9 – “no genes were significantly differentially expressed between the two lines at day 60.” This is a bit surprising given the confusion from supplementary figure 8 where some samples did not express CTIP2. Even examining supplementary figure 3, the heatmaps show some differences between KO1 and KO2. Unless this sentence is in relation to the genes associated with SZ?

Response: It must be remembered that, for each gene, the heatmap only plots its average expression level across all samples. The differential expression analysis also takes into account variation between samples and the number of genes tested in order to identify those genes with robust differences between lines. Our analysis identified no genes that were significantly differentially expressed between KO lines on day 60, meaning that none of the differences visible in the heatmap were statistically significant (i.e. had $P < 0.05$ after correcting for 13,855 tests) once within-line variability in expression was taken into account. Regarding the CTIP2 data, this measures protein rather than mRNA expression, which - although related - can show marked differences. It should also be noted that CTIP2 is only a single gene/protein – had we measured protein levels for all expressed genes, compared between lines and corrected for the number of genes tested (as we do when performing differential expression analysis), then CTIP2 may not have reached the threshold required for significance. This is not intended to downplay the variation noted in Supplementary Figure 8, but simply to make the difference between this and the differential expression analysis clear. The question originally posed by reviewer#3 was whether genes differentially expressed between KO lines were enriched for SZ common variant association: the analyses presented in our response show that this is not the case.